# Deformation recorded in polyhalite from evaporite detachments revealed by $^{40}$Ar/$^{39}$Ar dating

Lachlan Richards[1], Fred Jourdan[2], Alan Stephen Collins[1], Rosalind Clare King, [1]

[1]Tectonics and Earth Systems Group, Department of Earth Sciences, The University of Adelaide, SA 5005, Australia
[2] School of Earth and Planetary Sciences & JDL Centre, Curtin University, Perth WA 6845, Australia

*Correspondence to*: Lachlan Richards (lachlan.richards.08@aberdeen.ac.uk)

## Abstract

The Salt Range Formation is an extensive evaporite sequence in northern Pakistan that has acted as the primary detachment accommodating Himalayan orogenic deformation from the north. This rheologically weak formation forms a mylonite in the Khewra mines, where it accommodates approximate 40 km displacement and is comprised of intercalated halite and potash salts and gypsiferous marls. Polyhalite [$K_2Ca_2Mg(SO_4)4\cdot2H_2O$] grains taken from potash marl and crystalline halite samples are used as geochronometers to date the formation and identify the closure temperature of the mineral polyhalite using the $^{40}$Ar/$^{39}$Ar step heating laser and furnace methods. The diffusion characteristics measured for two samples of polyhalite are diffusivity ($D_0$), activation energy (Ea), and %$^{39}$Ar. These values correspond to a closure temperature of ca. 254 and 277 °C for a cooling rate of 10 °C/Ma. $^{40}$Ar/$^{39}$Ar age results for both samples did not return any reliable crystallization age. This is not unexpected as polyhalite is prone to $^{40}$Ar* diffusion loss and the evaporites have experienced numerous phases of deformation resetting the closed K/Ar system. An oldest minimum heating step age of ~514 Ma from sample 06-3.1 corresponds relatively well to the established early Cambrian age of the formation. Samples 05-P2 and 05-W2 have measured step ages and represent a deformation event that partially reset the K/Ar system based on oldest significant ages between ca. 381 Ma and 415 Ma. We interpret the youngest measured step ages, between ca. 286 Ma and 292 Ma, to represent the maximum age of deformation-induced recrystallisation. Both the youngest and oldest measured step ages for Samples 05-P2 and 05-W2 occur within the time of a major unconformity in the area. These dates may reflect partial resetting of the K/Ar system from meteoric water infiltration and recrystallisation during this non-depositional time. Else, they may result from mixing of Ar derived by radiogenic decay after Cambrian precipitation with partially reset Ar from pervasive Cenozoic deformation and physical recrystallisation.

## 1. Introduction

Diagenesis of evaporites from marine brines is initiated by the precipitation of specific minerals in sequence based on the composition of the parent brine with increasing salinity. Initially carbonates [$CaCO_3$] precipitate, followed by gypsum [$CaSO_4$] in penesaline brines, halite [$NaCl$] in supersaline brines and eventually bitter salts (K-Mg-salts) (Warren, 2006). Bittern salt

precipitation is a complex paragenetic process whereby the evolving brine chemistry and precipitate solubility define the terminal assemblage with any hydrological influx causing back reactions and alteration during or post lithification (Hardie, 1984 & 1990; Warren, 2006). The sedimentation of laterally extensive and thick evaporite deposits require hyper-arid climates with extreme evaporation, tectonically isolated basins with optimal hydrogeology to restrict brine refreshing and dissolution (Warren, 2006).

An ancient example of a thick, laterally extensive evaporite containing significant quantities of bittern salts is the Salt Range Formation in northern Pakistan (Jaumé and Lillie, 1988; Richards et al. 2015). It is comprised of a thick crystalline halite intercalated with bands of potash marl overlain by a gypsiferous marl and gypsum-dolomite (Ghazi et al., 2012). The Salt Range Formation acts as a detachment horizon for the distal foreland fold-thrust belt of the South Potwar Basin, which is a being driven by Himalayan orogenic deformation (far field stresses) and gravity gliding (near field stresses) (Jaumé and Lillie

1988, Davis and Lillie 1994, Richards et al. 2015). Ages of the Salt Range Formation are poorly constrained with trilobite trace fossils in the directly overlying Khewra Sandstone establishing the upper boundary as early Cambrian and the Precambrian metasedimentary basement rocks of the Indian Shield forming the lower boundary (Gee, 1989; Khan et al., 1986; Schindewolf and Seilacher, 1955).

Polyhalite is a bittern salt, forming as both a primary precipitate, but more commonly as a diagenetic secondary phase during

back reaction of gypsum with a K-Mg-$SO_4$ brine (Hardie, 1990; Warren, 2006). It is one of the primary potash (K-bearing) salts applicable to K/Ar, or its derivative, $^{40}Ar/^{39}Ar$ geochronology (e.g., Leitner et al. 2014). Since the initial investigation of $^{40}Ar$ abundance in K-bearing evaporite minerals (Aldrich and Nier, 1948) several studies have applied $^{40}Ar/^{39}Ar$ dating to evaporite minerals. Both K-Ar and $^{40}Ar/^{39}Ar$ dating of Miocene samples of polyhalite, kainite, and langbeinite from the Carpathian Foredeep Basin have successfully been used to determine depositional ages and recrystallisation ages after major

tectonic events (Leost et al., 2001; Wójtowicz et al., 2003). Similarly, polyhalite and langbeinite from the Castile and Salado Formations in southeast New Mexico, respectively, have been used with some success yielding ages of deposition and deformation (Brookins et al., 1980; Renne et al., 2001). While langbeinite is a more robust potash mineral, being less susceptible to Ar diffusion (Reiners et al., 2017), polyhalite is potentially a useful geochronometer with dates recovered from diagenetic and deformed polyhalite samples from the Haselgebirge Formation, a major evaporite detachment in the Northern

Calcareous Alps (Leitner et al., 2014).

In this study, we use the $^{40}Ar/^{39}Ar$ step heating process in an attempt to date grains of polyhalite from two samples from the Khewra Mines in the Salt Range, Pakistan. We also establish the Ar diffusion parameters and associated closure temperature of polyhalite in the K/Ar isotopic system. Though these results are semi-quantitative, they are contextualised with the structural history of the host formation to form a speculative interpretation of the deformation history.

 **2. Geological Background**

**2.1 Location**

Northward convergence of the Indian plate with the Eurasian plates in the northern regions of Pakistan and India has resulted in the classic continent-continent collision deformation structures on display in this region (Fig. 1A) (Jaswal et al., 1997). Continued convergence and crustal shortening since the Late Cretaceous saw the overthrusting of parts of the Indian passive margin over the Indian craton to the south, initiating and migrating new faults southward (Powell and Conaghan, 1973; Molnar and Tapponnier, 1977). The Main Mantle Thrust (MMT) separates the High Himalayas from the Lesser Himalaya to the south, which is itself separated from the Siwalik Hills and North Potwar Deformation Zone (NPDZ) by the Main Boundary Thrust (MBT) in northern Pakistan (Jaswal et al., 1997; Powell and Conaghan 1973) (Fig. 1).

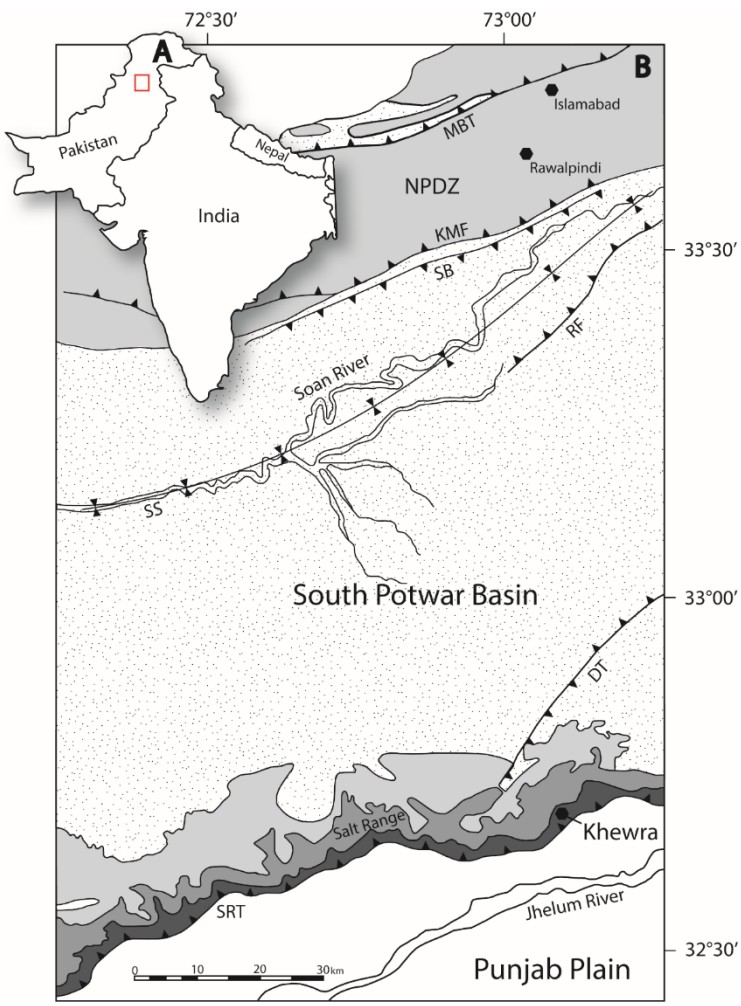


**Fig. 1. A) Location map of the study area. B) Geological map of the eastern Salt Range and Potwar Plateau. MBT = Main Boundary Thrust, NPDZ = North Potwar Deformation Zone, KMF = Khari Murat Fault, SB = Soan Backthrust, SS = Soan Syncline, RF = Riwat Fault, DT = Domeli Thrust, SRT = Salt Range Thrust. The colour shading references the geological Groups/Formations seen in Fig. 2 (after Richards et al., 2015).**

Further south, the NPDZ is separated from sedimentary rocks of the South Potwar Basin, also referred to as the Potwar Plateau, by the Khari Murat Fault (KMF) and Soan Backthrust (SB) in Fig. 1. The Potwar Plateau is a distal foreland fold-thrust-belt (FTB) that is thrust southward over a thick evaporite detachment, the Salt Range Formation (Davis and Lillie, 1994; Jaume and Lillie, 1988). At the southern extent of the Potwar Plateau, the Salt Range Thrust (SRT) has displaced Precambrian to

Eocene sedimentary rocks over Quaternary sediments of the Punjab Plain (Jaswal, 1997; Yeats et al., 1984). The SRT has allowed the southward transposition of nearly undeformed overburden and resulted in a critical taper wedge with a frontal angle of <1˚ (Jaume and Lillie, 1988). The Salt Range forms the southerly expression of the Himalayan orogeny. The range results from thrust ramping over a pre-existing basement normal fault, driven by a combination of both near (gravity gliding) and far field (continent continent collision) stresses (Davis and Lillie, 1994; Jaume and Lillie, 1988; Lillie et al., 1987). Recent

seismic activity indicates that the Salt Range Thrust is active, and moves at a rate of 3mm/yr, though slip is typically aseismic owing to the rheological weakness of the basal evaporites (Haq et al., 2013; Satyabala et al., 2012).

## 2.2 Stratigraphy

The oldest rocks in this region are the Precambrian crystalline basement of the Indian Shield with the nearest exposure in the Kirana Hills, 80 km south of the Salt Range (Fig. 2A) (Gee, 1989). These are unconformably overlain by evaporites of the Salt

Range Formation that formed in a restricted basin environment (Jaswal et al., 1997). The Salt Range Formation has three members: the Billianwala Salt, which is comprised of massive crystalline halite and sometimes banded with layers of potash marl, the Sahwal Marl, which is a red marl with some gypsum, and the Bandarkas Gypsum, which is a red marl containing both crystalline and non-crystalline folded and sheared gypsum (Fig. 2B)(Richards et al., 2015).

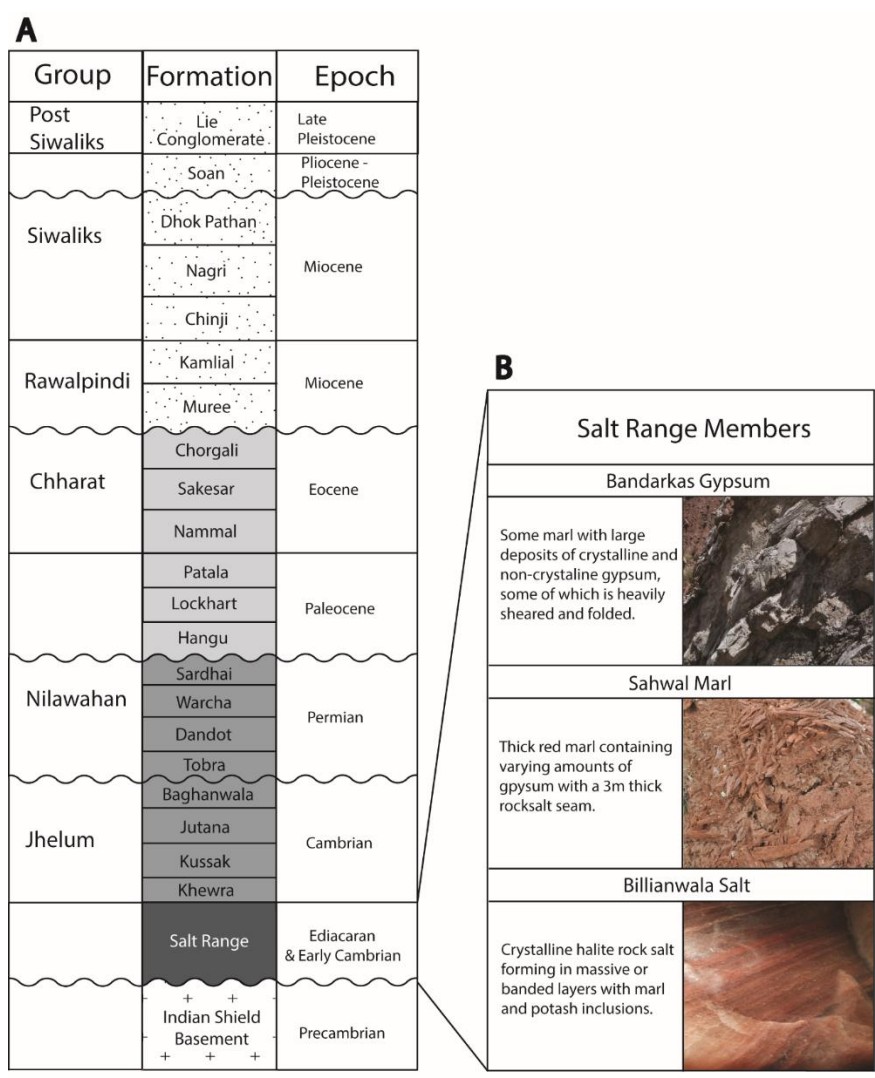

**Fig. 2. A) Stratigraphic column of the units within the study area B) Salt Range Member subdivisions (after Richards et al., 2015)**

Conformably above this are Cambrian marine sedimentary rocks of the Jhelum Group, predominantly consisting of maroon fine-grained sandstones and shales (Ghazi et al., 2012). These are first unconformably overlain by Permian tillites, sandstones, siltstones, and shales of the Nilawahan Group (Khan et al., 1986), then by Paleocene to Eocene fossiliferous carbonates and shales (Ghazi et al., 2012). Miocene to Quaternary units of the Rawalpindi and Siwalik Groups form a six km thick syntectonic molasse resulting from erosion of the forming Himalayas (Fig. 2A) (Grelaud et al., 2003).

Two samples taken from the Billianwala Member of the Salt Range Formation were collected from a mine wall within Khewra Mine (Fig. 1B). A detailed compositional and structural analysis of these samples is presented in Richards et al. (2015).

Sample SRLR-05 was taken from the massive crystalline halite and consists of 95% pure halite (NaCl) with pink orange

inclusions and bands comprised of carnalite ($MgCl_2.KCl.6H_2O$) and polyhalite ($2CaSO_4.MgSO_4.K_2SO_4.H_2O$) (Fig. 3A).

Sample SRLR-06 was taken from a thick band of maroon coloured potash marl containing halite boudins adjacent (8 m) to

sample SRLR-05 (Fig. 3B). Predominantly composed of marl, halite, gypsum, potash salts and clay minerals, the sample is

highly deformed showing mylonitic fabrics, boudinaged halite and, evidence of partial and total recrystallisation (Fig. 3B)

(Richards et al., 2015). As above, polyhalite and carnallite are the dominant potash salt.

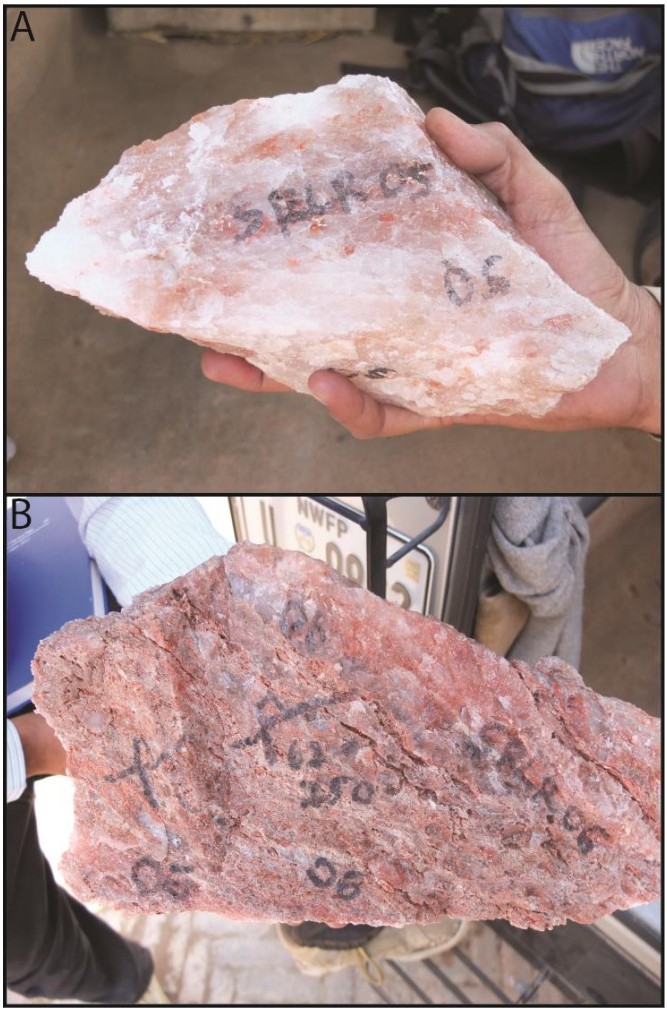

**Fig. 3. A) Hand sample SRLR-05 of halite (white) with some bittern salt inclusions (orange) from Khewra Mine. B) Hand sample SRLR-06 of potash marl (red orange) with clear to white crystalline halite from Khewra Mine. For a detailed mineralogical and structural analysis of these samples the reader is directed to Richards et al., 2015.**

## 3. Methodology

### 3.1 $^{40}$Ar/$^{39}$Ar Analysis

Polyhalite grains were separated by gently crushing the sample, then washing for 30 sec in distilled water to dissolve and remove any halite. Halite was carefully removed before irradiation as Cl produces interference $^{38}$Ar affecting the calculated ages (Esser et al., 1997; Leitner et al., 2014). As polyhalite is hydrous and only semi-soluble the grains do not experience alteration with such short contact when washed (Marcel et al., 2017). Grains were then sieved through multiple mesh sizes (500µm, 250µm, 100µm) to standardise the grain sizes; grains with a diameter between 150−210 µm were used for further analysis. Once separated polyhalite single crystal grains were loaded into aluminium disks, 1.9cm diameter and 0.3cm depth, and bracketed by small wells containing neutron fluence monitors. One sample (SRLR06-2.2) underwent irradiation with the Fish Canyon sanidine (FCs) as a neutron fluence monitor (28.294 ± 0.037 Ma, 1σ error; Renne et al., 2011). All other grains were irradiated with GA−1550 biotite (99.738 ± 0.100 Ma; 1σ error; Renne et al., 2011). The discs were Cd-shielded (to minimize undesirable nuclear interference reactions) and irradiated for 40 h in the Oregon TRIGA reactor in a central position. The mean J-value computed from standard grains within the small pits is 0.01082100 (±0.05 %) to 0.01086400 (±0.05 %), which is determined as the average and standard deviation of J-values of the small wells. Mass discrimination was monitored using an automated air pipette and provided a mean value of 1.003236 (±0.05 %) per dalton relative to an air ratio of 298.56 ± 0.31 (Lee et al., 2006). The correction factors for interfering isotopes were ($^{39}$Ar/$^{37}$Ar)Ca = 7.60 × 10$^{-4}$ (±1.2 %), ($^{36}$Ar/$^{37}$Ar)Ca = 2.70 × 10$^{-4}$ (±0.74 %) and ($^{40}$Ar/$^{39}$Ar)K = 7.30 × 10$^{-4}$ (±12.4 %). These $^{40}$Ar/$^{39}$Ar analyses were conducted at the Western Australian Argon Isotope Facility at Curtin University. The grains were step-heated using a 110 W Spectron Laser Systems, with a continuous Nd-YAG (Infrared; 1064 nm) laser rastered over the sample for 1 min to ensure that all the gas has been extracted and a homogenised temperature was reached across the samples. Contemporaneous step-heating experiments were run to determine the diffusion kinetics of polyhalite. Sample SRLR-05 and SRLR-06 underwent separate diffusion experiments with each sample represented by multi-crystal aliquots (10–20 grains) of roughly equant 150– 210 µm diameter crystals. These samples were placed inside copper foil packages before being transferred to the double vacuum high frequency Pond Engineering furnace and step-heated. A Pond Engineering thermocouple is used to measure extraction temperatures. Each extraction step is 10 minutes, which includes eight minutes of static temperature with first two minutes ramping to the desired temperature. Mass spectrometer analysis occurs at the end of each step where the temperature drops by 150 °C. For both gas extraction approaches, the gas was purified in a stainless-steel extraction line using one GP50 and two SAES AP10 getters. Argon isotopes were measured in static mode using a MAP 215-50 mass spectrometer (resolution of ~450; sensitivity of 4 × 10$^{-14}$ mol/V) with a Balzers SEV 217 electron multiplier. The data acquisition was performed with the Argus programme written by M.O. McWilliams and was run under a LabView environment. The raw data were processed using the ArArCALC software (Koppers, 2002) and ages were calculated using the decay constants recommended by Renne et al. (2011). $^{40}$Ar blanks range from 1 × 10$^{-16}$ to 2 × 10$^{-16}$ mol and were monitored every third step; calculated age data are presented with 2σ errors.

## 3.2 $^{40}$Ar/$^{39}$Ar Polyhalite Diffusion Calculations

We calculated the D values for our experiments using equation 5.29 in McDougall and Harrison (1999) using the fraction of $^{39}$Ar and duration of each step. For each of our samples, -Ln (D) vs. 10000/T values were plotted on Arrhenius plots (Fig. 7). Arrhenius law calculations describe the first order kinetic loss of a diffusant, in this case $^{39}$Ar, as a function of temperature (Dodson, 1973).

$$\ln\left(\frac{D}{a^2}\right) = \ln\left(\frac{D_0}{a^2}\right) + \left(\frac{-E_a}{R}\right)\left(\frac{1}{T}\right) \tag{1}$$

R is the gas constant, Ea is the activation energy, D is the diffusion coefficient, $D_0$ is the pre-exponential diffusion factor, a is the diffusion size, T is the temperature. The two diffusion parameters, Ea & $D_0$, are extracted from the array defined by the data presented in the Arrhenius plots up until temperatures where the crystals broke down and began to melt. We used a crystal radius of 90 ±15 µm and a spherical geometry for the calculation as this geometry is appropriate for all grain shapes, other than platy minerals, with little effect on the diffusion results (Blereau et al., 2019). Errors on the y-axis intercept $D_0$ and slope Ea were calculated using a robust regression (Isoplot v3.7; Ludwig, 2003) since the scatter on the regression line is much larger than the uncertainties on the individual measurements. Ginster and Reiners (2018) propose a range of error propagation solutions for deriving noble gas diffusion parameters, one of which is the non-weighted ordinary least-square regression that we employ in this study. As it is a non-weighted solution it does not require uncertainty to be calculated for each step. Additionally, we also calculated a weighted ordinary least-square regression considering the error in Ln(D/a$_2$) for a substantive comparison to be made between different approaches. Closure temperatures were calculated using the formulas presented in Dodson (1973) with a cooling rate of 10°C/Ma with uncertainties at the 2σ level (95% confidence).

## 4. Results

### 4.1 $^{40}$Ar/$^{39}$Ar dating

Nine polyhalite single crystal aliquots, two taken from sample SRLR-05 and seven taken from sample SRLR-06, underwent step-heating $^{40}$Ar/$^{39}$Ar dating. The uneven distribution between samples results from the strong compositional difference between the two samples; Sample SRLR-05 comprised of 95% halite so finding polyhalite crystals was difficult. A summary of results for all nine samples presented in Table 1.

| Sample | Steps | K/Ca | ± 2σ | Youngest Measured Step Age ± 2σ (Ma) | | % $^{39}$Ar | Oldest Measured Step Age ± 2σ (Ma) | | % $^{39}$Ar |
|---|---|---|---|---|---|---|---|---|---|
| 05-P2 | 9 | 0.82 | ±0.02 | 292 | ±1 | 24.4 | 381 | ± 1 | 8.5 |
| 05-W2 | 7 | 0.83 | ±0.02 | 286 | ±1 | 22.8 | 415 | ± 1 | 15.5 |
| 06-1.2 | 7 | 0.0017 | ±0.0002 | N/A | - | 17.0 | N/A | - | 14.3 |
| 06-2.1 | 8 | 0.82 | ±0.02 | 187 | ±1 | 4.7 | 348 | ± 1 | 29.9 |
| 06-2.2 | 7 | 0.0046 | ±0.0002 | N/A | - | 52.0 | N/A | - | 8.7 |
| 06-3.1 | 9 | 0.82 | ±0.02 | 470 | ±2 | 10.4 | 514 | ± 3 | 19.5 |
| 06-3.2 | 9 | 0.77 | ±0.02 | 272 | ±2 | 5.7 | 424 | ± 2 | 15.6 |
| 06-4.2 | 7 | 0.0185 | ±0.0006 | 62 | ±15 | 49.2 | 618 | ± 92 | 5.7 |
| 06-4.2' | 8 | 0.0209 | ±0.0005 | N/A | - | 36.2 | N/A | - | 4.5 |

**Table 1. Polyhalite $^{40}$Ar/$^{39}$Ar dating summary of results, Plateau Age, ±, MSWD, %$^{39}$Ar, and inverse isochron data are not presented as no useable plateau or isochron ages were determined. Note that the single step error ages are not related to any meaningful geological events but are rather semi-quantitative numbers indicating minimum and maximum ages.**

$^{40}$Ar/$^{39}$Ar geochronology results are presented with age spectra plots, presentation of these have been separated based on two populations identified by having relatively high or low K/Ca ratios (Fig. 4). For geological significance to be assigned to these results the observed ages and plateau ages should overlap within 2σ error confidence. Our criteria for a plateau age is at least three consecutive steps having overlapping measured ages and at least 70% of total $^{39}$Ar released from the sample, with mini-plateau ages containing between 50 and 70% of the total $^{39}$Ar released (Jourdan et al., 2020). As the calculated ages of all

samples only returned one plateau age, with significant errors, precise ages of the formation are unable to be derived. However, the qualitative information regarding the minimum age of crystallisation and maximum age of diffusion loss or recrystallization can be derived from the present results (see discussion by Jourdan, 2012). As such the oldest and youngest step ages are presented in Table. 1, with the oldest step age reflecting a minimum age of primary crystallization and the youngest step age reflecting the maximum age of deformation and Ar diffusion loss.

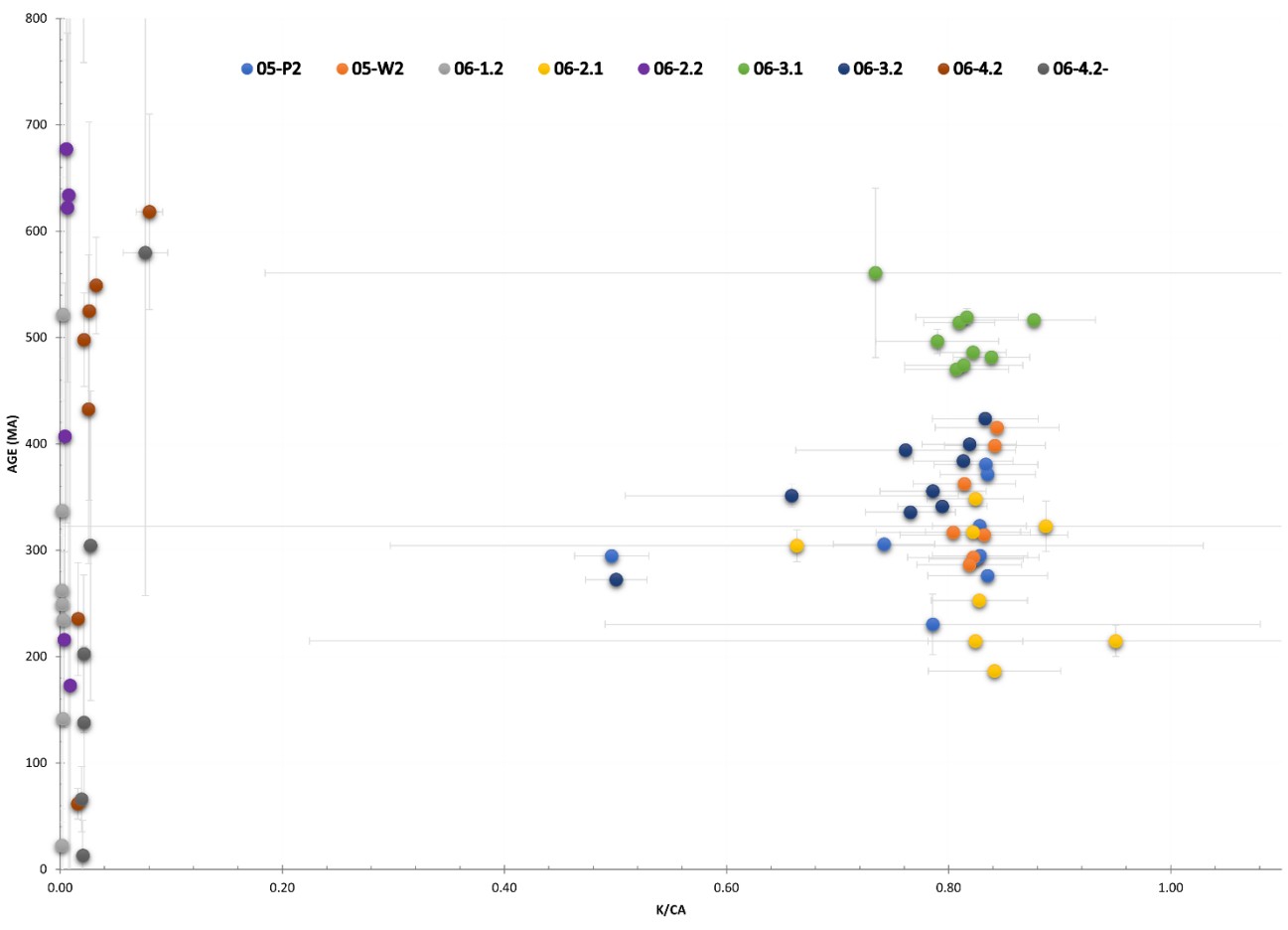

**Fig. 4. Measured age vs K/Ca plot displaying two distinct populations within the samples tested.**

Stacked, measured age and K/Ca spectrum, plots are presented below with Fig. 5 showing samples with relatively high K/Ca values and Fig. 6. With relatively low K/Ca values. All five plots in Fig. 5 show a stepped diffusion-like profile of increasing age with cumulative $^{39}$Ar released rather than an idealised plateau. The initial and final steps in the measured age plots for 05-P2, 05-W2, and 06-3.1 (Fig. 5 A, B, & D) converge around similar ages representing the youngest and oldest significant ages for their respective samples. This also occurs with plots for 06-2.1 and 06-3.2; however, the errors associated with these steps are high and as a result divert from the stepped diffusion profile at both start and end steps. All five K/Ca ratios form flat profiles with near identical ratios, 0.82–0.83, except for sample 06-3.2 at 0.77. This consistency is indicative of very strong compositional homogeneity between individual crystals. As these sample are all single crystal analyses it lends weight to our interpretation that Ar generation from these samples occurs from single diffusion domain. This notion is addressed further in the discussion.

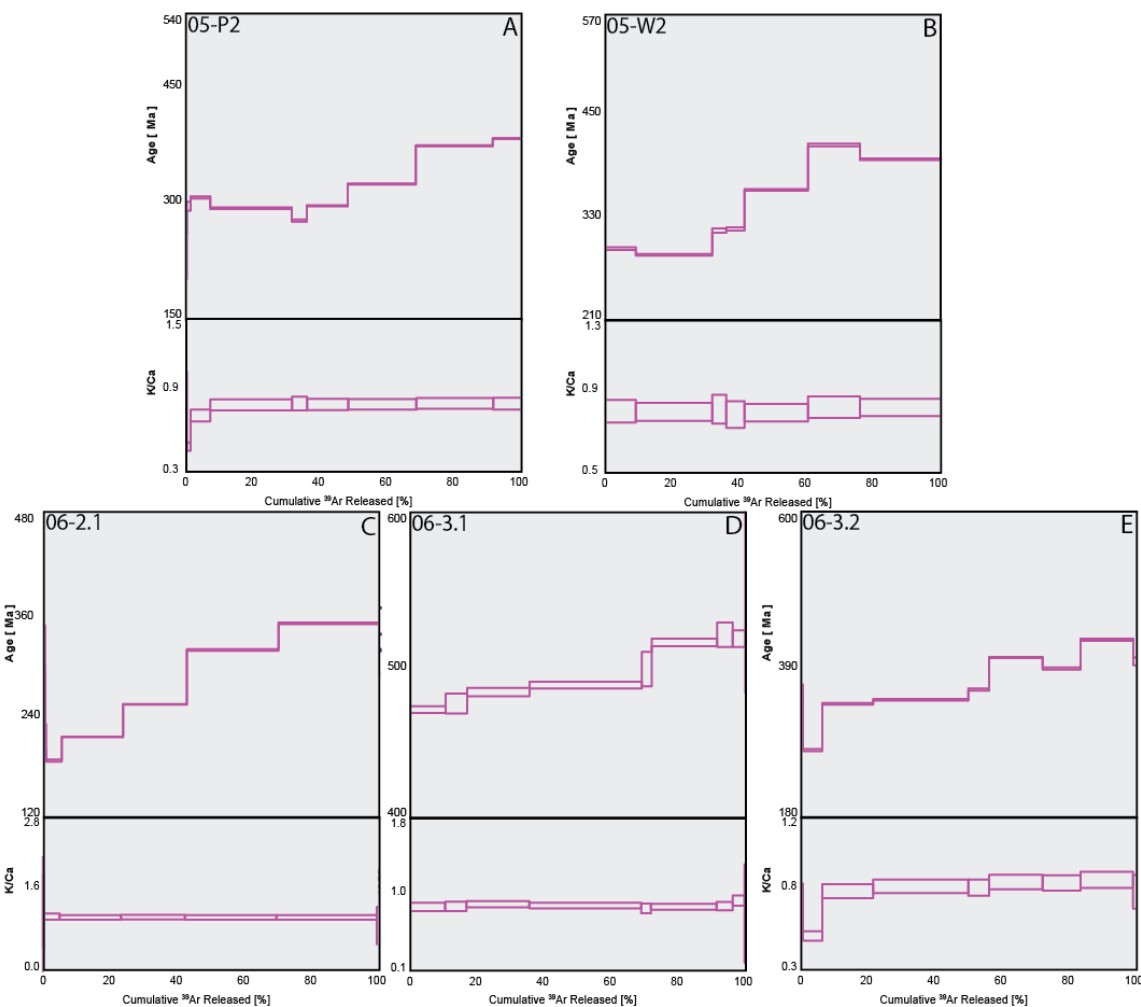

**Fig. 5. Combined age spectra and K/Ca plots for samples 05-P2 (A), 05-W2 (B), 06-2.1 (C), 06-3.1 (D), 06-3.2 (E). The thickness of individual spectra blocks is indicative of measurement uncertainty.**

Although sample 06-1.2 (Fig. 6A) shows a calculated plateau age, it produced very little gas, with low K/Ca values and analyses barely above blank levels, yielding very imprecise ages; the results from this sample have therefore been discarded. Similarly, only measured step ages with relatively small uncertainty are selected. For example, Sample 06-4.2' (Fig. 6D) has a youngest step age of 160 ± 1330 Ma and oldest step age of 204 ± 754 Ma; neither of these ages are precise enough to be useful so are disregarded and the nearest step age of sufficient precision is selected. All four plots in Fig. 6 show a stepped diffusion profile of increasing age with cumulative $^{39}$Ar released. An interesting trend is observed in plots 06-2.2, 06-4.2, and 06-4.2' showing sequential large jumps in step ages, along with measurement uncertainty, beginning from ~60% cumulative Ar released (Fig. 6 B-D). A cause for this may be mineral breakdown and dehydroxylation, which is covered in detail in the

discussion. Most notably, the analytical error in the data for these samples are extremely large, with the smallest error at 2σ being half the measured age value, 06-4.2': 13 ± 33 Ma, and the largest being greater than three times the measured age value, 06-1.2: 1071 Ma ± 3.3 Ga (Table. 1). As a result of such imprecision, no interpretations can be drawn from these data. The

220 K/Ca ratios in all four plots of Fig. 6 are barely above zero for the majority of argon release steps until ramping sharply at ~90% cumulative argon released. This behaviour is highly inconsistent with the relatively straight K/Ca profiles observed in Fig. 5 and combined with the imprecise step ages indicate these samples are unlikely to be polyhalite, but rather another mineral with low potassium.

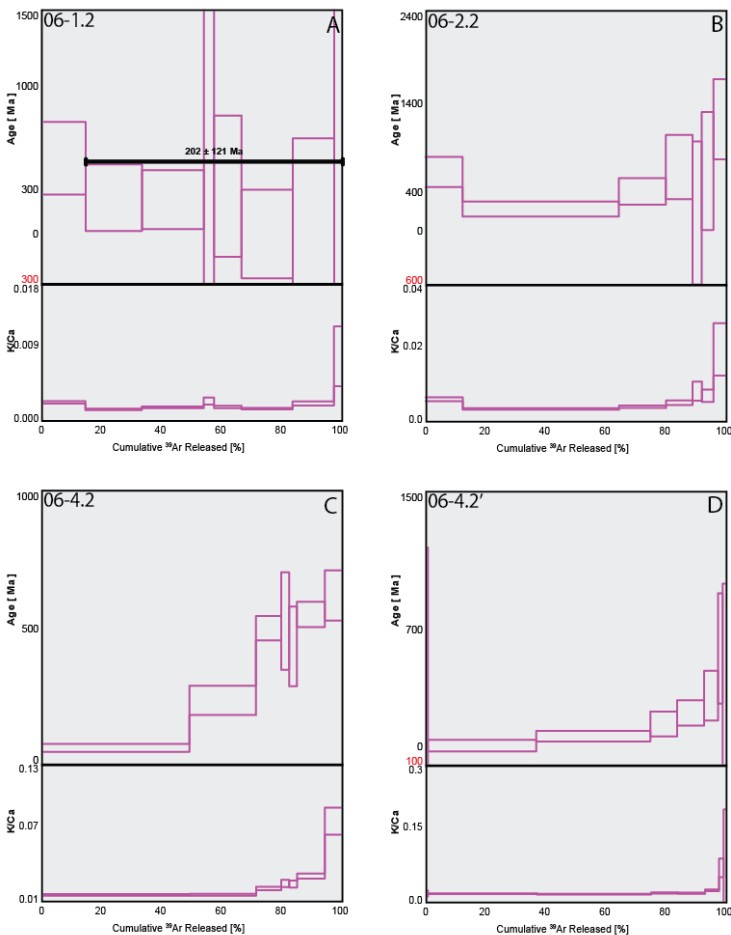

**Fig. 6. Combined age spectra and K/Ca plots for samples 06-1.2 (A), 06-2.2 (B), 06-4.2 (C), 06-4.2'(D). The thickness of individual spectra blocks is indicative of measurement uncertainty.**

Typically, inverse isochron diagrams are created as a secondary age calculation and to assist with measuring the value of trapped $^{40}Ar/^{36}Ar$ (McDougall and Harrison, 1999). Isochron plots of these data are not presented here as they do not add any additional value as most data points plot along the X-axis and do not form a mixing line to allow for age calculation. Although

it is perhaps not common practice to extensively discuss strongly perturbed age spectra; we do so here to derive as much as semi-quantitative information as we reasonably could, in part due to the limited $^{40}$Ar/$^{39}$Ar geochronology data on polyhalite in the literature.

## 4.2 Closure Temperature

Separate step-heating experiments were performed to determine the closure temperature of Ar in polyhalite. The results of 235 which are presented below as Arrhenius plots measuring diffusion of $^{39}$Ar for samples SRLR05 (Fig. 7 A) and SRLR06 (Fig. 7 B & C).

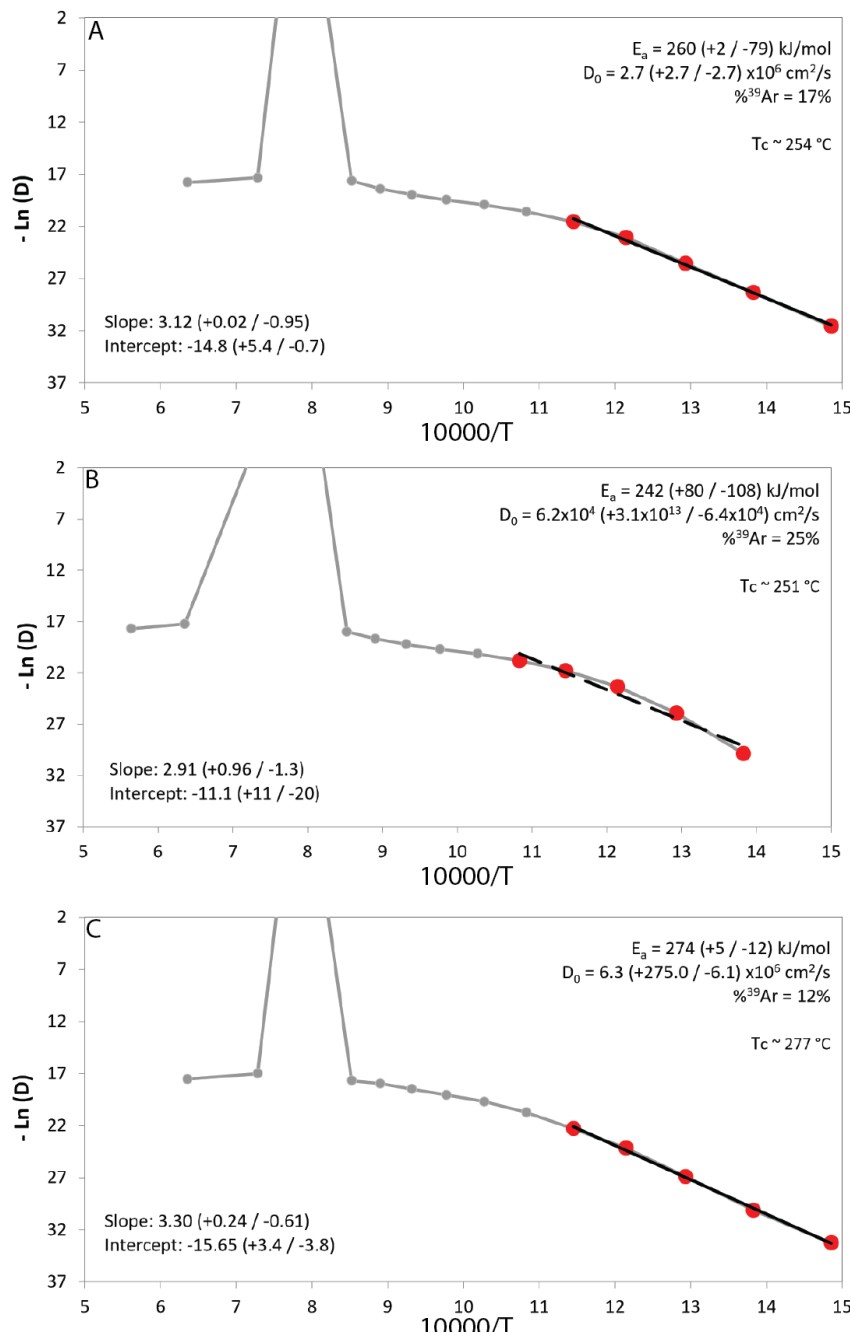

**Fig. 7. Arrhenius plots of argon diffusion coefficients calculated from step-heating experiment data against a reciprocal absolute temperature of multigrain polyhalite aliquots from two sample SRLR05 (A) and SRLR06 (B &C). The red dots are the steps used for the regression line (black line) from which the values of $E_a$, $D_0$ and $T_c$ were calculated.**

The Arrhenius plots (Fig. 7) indicate $^{39}$Ar gas release from the polyhalite grains occurred in two distinct stages. The first is depicted as the straight array of points at low temperatures, from which the regression lines and diffusion properties of three multi-crystal aliquot populations are calculated. This low temperature stage exhibits a steady and relatively rapid release of Ar, between 12–25% of total trapped Ar, with a moderate slope (Fig. 7). This indicates a consistent diffusion profile incorporating little to no Ar release from mineral defects, cracks, or cation sites of low K retentivity. This diffusion profile is also consistent with single domain degassing behaviour (McDougal and Harrison, 1999). Calculations based on our Arrhenius plot (Fig. 7) data arrays yielded $D_0$ and $E_a$ values between ~2.71 x $10^6$ – ~6.29 x $10^6$ cm$^2$/s and ~260 – ~274 kJ/mol respectively (Table. 2). From these values we calculated closure temperatures ($T_c$) ranging from ~254 – ~277 °C using a crystal radius of 90 ±15 µm and a standard cooling rate of 10°C / Ma (Table. 2).

The second stage is defined by a curved array at higher temperatures that we interpret is a result of progressive mineral transformation, dehydroxylation, and break down; this is explored in the discussion below. Additionally, there is a spike at step 8 on all three Arrhenius plots (Fig. 7), also likely caused by mineral break down. We note that the regression line, and subsequently calculated diffusion properties, in Fig. 7B do not fit the data points as well as the two other samples as a result of a slight curvature in the data array even at low temperatures. Yet, diffusion properties and resulting $T_c$ still closely matches the other results. While it is important to note that these calculated diffusion properties are not fully accurate due to the possible dihydroxylation breakdown of polyhalite during step-heating, it supports our very conservative approach concerning uncertainty propagation. As described in the methods we present three distinct regression calculations for each plot (Table. 2); this is done to demonstrate the differences between standard uncertainty propagation of Ginster and Reiners (2018) and the calculated $D_0$ and $E_a$ uncertainty arising from slope deviation in Isoplot. If error propagation is similar across all steps in the calculations there should be no difference between the models, as such it is reassuring that all our calculations show only slight differences. Rather than quoting the wider range of values between all three approaches elsewhere in this paper we use the ISO values (Table. 2) as they best represent the real geological uncertainty and at least partially offset the problem of measuring diffusion parameters of hydrous minerals (Harrison et al., 2009).

| Sample / Run Number | Regression method | $D_0$ (cm²/s) | $E_a$ (kJ/mol) | [39]Ar lost (%) | $T_c$(°C) |
|---|---|---|---|---|---|
| **SRLR05 / Poly1** | ISO | $2.7 (+2.7 / -2.7) \times 10^6$ | $260 (+2 / -79)$ | 17 | **~254** |
| | OLS | $1.9 (\pm5.9) \times 10^5$ | $244 \pm 20$ | - | **~242** |
| | WLS | $3.0 (\pm2.1) \times 10^5$ | $260 \pm 2$ | - | **~252** |
| **SRLR06-1 / poly 2** | ISO | $6.2 \times 10^4 (+3.1 \times 10^{13} / -6.4 \times 10^4)$ | $242 (+80 / -108)$ | 25 | **~251** |
| | OLS | $4.0 \times 10^4 (\pm36.7 \times 10^4)$ | $239 (\pm62)$ | - | **~245** |
| | WLS | $2.1 \times 10^3 (\pm27.9 \times 10^3)$ | $212 (\pm90)$ | - | **~213** |
| **SRLR06-2 / poly3** | ISO | $6.3 \times 10^6 (+275.0 \times 10^6 / -6.1 \times 10^6)$ | $274 (+5 / -12)$ | 12 | **~277** |
| | OLS | $3.5 \times 10^6 (\pm8.4 \times 10^6)$ | $271 (\pm14)$ | - | **~271** |
| | WLS | $7.7 \times 10^8 (\pm106.0 \times 10^8)$ | $305 (\pm92)$ | - | **~289** |

**Table. 2 Results of step-heating diffusion experiments. Regression acronyms ISO = robust regression in Isoplot V3.7 (Ludwig, 2003), OLS = non-weighted ordinary least square, WLS = weighted (error in $\ln(D/a^2)$) least square.**

## 5. Discussion

The purpose of this work is to investigate $^{40}Ar/^{39}Ar$ geochronology to determine a depositional or resetting age for the polyhalite in the Salt Range Formation as well as establishing the closure temperature for the mineral polyhalite. The Salt Range Formation has an upper age constraint of early Cambrian, from trilobite and brachiopod fossils in the conformably overlaying Khewra Formation. It also rests unconformably over the Precambrian basement of the Indian Shield (Khan et al., 1986). Our step heating laser experiments did not return plateau ages, the one sample that did return a plateau age is invalid due to unacceptably high errors.

Of the nine samples that underwent step-heating analysis, six were determined to be of insufficient quality for any interpretation. Samples 06-1.2, 06-2.2, 06-4.2, and 06-4.2' contained significant analytical errors. These samples coincide with the 2nd population of grains in Fig. 4 with K/Ca ratios much lower than the other samples. We suspect the analytical errors associated with these samples may correspond to the overall low K values. Samples 06-2.1 and 06-3.2 also exhibit step-heating profiles typical of single domain diffusion (but see discussion below). The lack of convergence of early and late steps to a measured age of any significance results in vaguely interpretable spectral plots of unknown significance (Fig. 5). Observations of halite boudins within the mylonitic potash marl, from which the '06' samples were taken, suggest that the marl layers may have acted as high-strain fluid flow pathways within the much thicker massive crystalline halite units (Richards et al., 2015).

These bands of potash marl have the highest percentage K-Mg salts within the formation (Fig. 3B); ultimately this results in a higher likelihood of dissolution, back reaction, altered brine chemistry, and recrystallisation (Warren, 2006). This process of dissolution and recrystallisation may partially or wholly reset the K–Ar isotopic system by untrapping the daughter decay products; consequently, the age calculations based on isotope ratios will reflect the younger deformation / recrystallisation age or an incorrectly calculated age between the depositional and recrystallisation ages if dissolution is partial (Jourdan, 2012; McDougall and Harrison, 1999).

## 5.1 $^{40}$Ar/$^{39}$Ar systematics

While it is impossible to determine the initial deposition age of an evaporite sequence with no known lower boundary without a plateau age from $^{40}$Ar/$^{39}$Ar geochronology, we can derive a minimum age of crystallization based on the oldest measured step age from the last heating step. Conversely, the youngest steps will provide a maximum age for secondary, tertiary, or further, perturbation event such as an heating event or fluid-induced recrystallization. Intermediate step ages will effectively be a mixture between the initial crystallization and later alteration events.

*Formation of the first generation of polyhalite*

Take for example, sample 06-3.1 (Fig. 5) that returned an oldest measured step age of ~514 Ma; we attribute this age to represent a minimum age, probably not so far from the time at which this polyhalite grain precipitated. This age may represent the depositional age of precipitation from a surface brine; however, we believe it is more representative of precipitation during lithification as polyhalite is most commonly a secondary evaporite occurring as a back reaction (Warren, 2006).

The oldest measured step ages for samples 05-P2 and 05-W2 (~381 Ma and ~415 Ma respectively) are significantly lower than the youngest age of the Salt Range Formation of ca. 514 Ma from our step heating experiment on sample 06-3.1 or from the stratigraphically constrained early Cambrian age (Table. 1). As such, these samples must have experienced conditions capable of significantly resetting the K/Ar decay system. For the majority of minerals this occurs when the mineral is heated beyond its closure temperature. However, as polyhalite is a chemical precipitate, percolating fluids of the correct composition are capable of dissolving and re-precipitating new minerals (Warren, 2006). As polyhalite forms from precipitation or alteration rather than magmatic crystallisation it is most likely these minerals formed well below the closure temperature, effectively locking both K and Ar with insignificant post-formation diffusion. For these samples, whether alteration is thermally derived or purely recrystallisation, we can establish that the oldest measured step ages represent a minimum age at which these new polyhalite grains first formed.

*Alteration / recrystallisation age*

Considering again these two samples: 05-P2 and 05-W2, we constrain the age of termination of the most recent alteration processes by the youngest significant ages. These youngest significant ages — ~286 Ma for sample 05-W2 and ~292 Ma for

sample 05-P2 — provide a maximum age for the recrystallization of at least a second generation of polyhalite, or for an heating event that partially reset the first generation of polyhalite (Table. 1). Placed in geological context these dates correspond with the time of the unconformity between the middle Cambrian sequence of the Baganwalla Formation and the early Permian glacio-fluvial to shallow marine sediments of the Tobra Foramtion (Khan and Khan, 1979; Khan et al., 1986). It is tempting to suggest that these ages reflect recrystallisation by circulating meteoric fluids during this non-depositional time, possibly

with glacial meteoric water having infiltrated the evaporites around the time of rifting and break-up of the northern Gondwana border. While possible, this interpretation is less likely than the following scenarios. Collision between the Indian and Eurasian plates is currently ongoing with crustal movement rates estimated to be ~3 mm/y (Satyabala et al. 2012). Numerous <4.9 Mw earthquakes in the region and neotectonic features within the Salt Range Thrust indicating that the location where these samples were collected is currently active (Haq et al. 2013). Combining this information with the youngest calculated step age of ~62

325 Ma, and the observation of the extreme microscopic deformation in the mylonite (Richards et al. 2015), we can speculate on the nature of the interaction between the Ar system and recrystallization. We purpose the following scenarios, any of which may occur individually or concurrently:

1. Stress-induced recrystallisation experienced by these samples is only ever partial, with intracrystalline domains preserved that remain unrecrystallised, else only the most recent deformation events to reset the system should be
recorded in the Ar system.

2. If total recrystallisation has occurred, but the Ar system isn't fully reset, then recrystallisation retains Ar despite the host crystal structure's recrystallisation. This would suggest little to no fluid is present during recrystallisation.

3. Deformation events resulting in deformation and recrystallisation may have heterogeneously affected the Salt Range Formation with some grains (e.g. those in boudins), preserving different microstructural and isotopic records.

4. The samples analysed were only recrystallised at this age while others may have experienced more recent deformation resetting its closed Ar system but were not analysed.

*Causes of intermediate step ages*

Fitting with most published $^{40}Ar/^{39}Ar$ dating studies of potash salts, the results presented in this study display significantly
younger ages of formation than their known upper limit of base Cambrian age. We suggest that later deformation events are the primary cause of this open system behaviour, rather than a result of prolonged thermally induced diffusion. Located within a tectonically active setting, with evidence for recent (0.4 – 2.1 Ma) to currently active movement nearby, and hosted in or close to evaporate mylonites, these samples have experienced, at least partial, grain boundary migration and re-crystallisation (Yeats et al., 1984; Jaswal et al., 1997; Haq et al., 2013). However, microstructural work by Richards (2021) ascertains that
even in heavily deformed evaporites, earlier microstructures are preserved, suggesting that these intracrystal domains may retain radiogenic Ar and preserve older ages.

## 5.2 Diffusion characteristics of polyhalite

Our step heating furnace experiments to determine the diffusion parameters for polyhalite have resulted in $E_a$ between ~260 – ~274 kJ/mol and $D_0$ between ~2.71 x $10^6$ – 6.29 x $10^6$ $Cm^2$/s. Linear regression of these parameters returned in calculated closure temperatures ($T_c$) between ~254 – ~277 °C for a cooling rate of 10°C/Ma. Langbeinite, the other K-bearing salt for which diffusion characteristics have been calculated, has comparatively lower values; Ea from 178 – 184 kJ/mol, $D_0$ at 1.0 x $10^{31}$ $Cm^2$/s, and $T_c$ at 200 °C (Lippolt & Oesterle, 1977; Renne et al., 2001). A more detailed list of argon diffusion characteristics can be found in Baxter (2010).

Our diffusion properties (Table. 2) have been derived from the dominant array of data points (Fig. 7 Red dots) that ideally reflects an homogenous single domain mineralogy. We assert single domain diffusion is most likely for these experiments and substantiate our interpretation with a few observations. Firstly, five single crystal analyses from our laser experiments (Fig. 5) indicate high and consistent K/Ca and ratios. As mentioned before, this suggests homogeneous composition and consistent mineralogy between individual crystals. This alone does not suggest single domain diffusion as a flat K/Ca ratio can also be achieved if the measured grains contain intra crystalline domains of differing sizes as long as they are compositionally homogeneous. Secondly, these samples contain a high percentage of radiogenic argon indicating the grains are resistant to atmospheric argon inclusion. Thirdly, this is further evidenced by the alignment of points on the Arrhenius plots (Fig. 7) that show steady Ar diffusion at a moderate slope. All three Arrhenius plots show no signs of low temperature rapid release of low percentage volume of Ar at shallow slopes that is often interpreted as fast release argon from cracks and defects, which would become a separate diffusion domain (Blereau et al., 2019; Thern et al., 2020).

## 5.3 Thermal stability and dehydroxylation of polyhalite

Dehydroxylation in a mineral occurs during heating above a mineral specific temperature resulting in the loss of the hydroxyl group (OH). This phenomenon is pertinent to incremental heating experiments involving hydrous minerals, which may undergo irreversible structural and morphological phase changes inherently altering the active diffusion mechanism and subsequently derived kinetics (Cassata and Renee, 2013; Gaber et al., 1988; Harrison et al., 2009; Lee et al., 1991; Thern et al., 2020). As such, volume diffusion kinetics derived from minerals undergoing phase changes may be measuring the original mineral, a new phase or mineral, or a composite of both depending on when this phase change occurs.

Polyhalite is known to dehydrate with the following first order reaction between 280 ˚C to 360 ˚C at pressures between 0.5 and 6.1 bars from heating experiments (Nathans, 1963).

$$K_2Ca_2Mg(SO_4)_4 \cdot 2H_2O \rightarrow K_2CaMg(SO_4)_3 + CaSO_4 + 2H_2O \qquad (2)$$

Further, thermal experiments by Wollmann et al. (2008) identified the dehydration characteristics of polyhalite and its analogues (polyhalite with cation replacements Mn, Fe, Co, Ni, Zn, & leightonite) with dehydration onset at 255 ˚C and

peaking at 343 °C for polyhalite and onset between 185 – 311 °C for the various analogues. The specific thermal decomposition

reactions observed by Fischer et al. (1996) and expanded upon by Xu et al. (2016) show polyhalite dehydrates into anhydrite, two solid solution langbeinite-type phases with different Ca/Mg ratios, and water vapour between 237 – 343 °C. This is accompanied by a 5.8% mass loss associated with water vapour removal (Xu et al., 2016).

$$K_2Ca_2Mg(SO_4)_4 \cdot 2H_2O \rightarrow K_2Ca_xMg_{2-x}(SO_4)_3 + K_2Ca_yMg_{2-y}(SO_4)_3 + CaSO_4 + 2H_2O \qquad (3)$$


Upon heating to 646 °C the two langbeinite phases combine to a single-phase triple salt (Xu et al, 2016).

$$K_2Ca_xMg_{2-x}(SO_4)_3 + K_2Ca_yMg_{2-y}(SO_4)_3 \rightarrow K_2CaMg(SO_4)_3 \qquad (4)$$

The polyhalite unit cell parameters have been established by Wollman et al. (2008) and the variation of these as a function of temperature by Xu et al. (2016). These thermal experiments are conducted at atmospheric conditions or with variations to pressure so it is unknown whether polyhalite decomposition will differ from these results as minerals have displayed lower phase transformation temperatures in vacuo (Vasconcelos et al., 1994c). While the combined effect of geological conditions (temperature, lithospheric pressure, water volume) on these reactions is unknown, the dihydroxylation temperature of

polyhalite presented here, 237 – 343 °C, is around our calculated $T_c$ 254 – 277 °C. This indicates polyhalite is Ar retentive below its phase transformation onset temperature in vacuo, which impedes accurate calculation of diffusion kinetics and closure temperatures with this method. As such, our calculated diffusion characteristics and closure temperatures are only semi-quantitative, representing a first attempt at measuring the diffusion kinetics of polyhalite. Conducting hydrothermal diffusion experiments would serve as a great alternative and may alleviate the dehydroxylation issue. Hydrothermal diffusion

experiments increase the thermal stability range of a mineral before decomposition onset allowing for higher analytical temperatures to be reached (e.g. Baldwin et al., 1990; Giletti, 1974; Harrison et al., 2009). Unfortunately, this approach was not possible as the facilities were unavailable during this study. It would make an ideal continuation of this work on expanding our understanding of the diffusion parameters of polyhalite and its applicability to future geochronological work.

**5.4 Implications for diffusion measurements**

The dehydroxylation of polyhalite into two langbeinite-type phases and subsequent breakdown into the triple salt ($K_2CaMg$($SO_4)_3$) has significant implications for interpreting and understanding our results. While the first dehydration reaction of polyhalite onsets at 237 °C and peak dehydration occurring at 343 °C it does not imply total phase transformation at this temperature. Both our laser and furnace experiments achieve temperatures well above the mineral breakdown point of our samples, so it is undeniable that these analyses document a combination of the above mineral phase transformation chain.

When considering what is being measured during our experiments a few hypotheses can be made.

- The first: polyhalite is Ar retentive well above its mineral decomposition temperature, so all Ar release is due to a combination of crystallographic reconstitution during phase change and langbeinite diffusion until high temperature diffusion through the triple salt.

- The second: Polyhalite releases Ar both before and during phase transformation during low temperature steps, once fully transformed Ar release occurs with langbeinite diffusion properties until high temperature diffusion through the triple salt.

If our calculated Tc is taken into consideration the second hypothesis is certainly the more likely. The complex behaviour we observe in our results is undoubtedly linked to this complicated degassing and phase transformation sequence.

We suspect the stepped age profiles (Fig. 5) and curved data array in higher temperature Arrhenius plots is a result of mineral breakdown through this phase sequence. The polyhalite analogues display a wide range of thermal dehydration onset (185 – 311 ˚C). As such, the increased curvature of the data array for SRLR06 / Poly2 (Fig. 7B) that did not allow for an accurate regression calculation may result from a compositional difference imparting earlier onset dehydroxylation. Since there is so little information regarding polyhalite and no established minimum $T_c$ (excluding our results) we suggest that our semi-

quantitative results for polyhalite Ar diffusion kinetics are conservatively accurate. Furthermore, knowing even imprecise values for diffusion properties and $T_c$ may prove useful and aid interpretations in the future.

## 6. Conclusions

The [40]Ar/[39]Ar step heating geochronology performed on these nine samples was partially successful. Despite being unable to

determine significant plateau ages to reliably date the evaporites of the Salt Range Formation, we can speculate on a few key points regarding the deformation history of these evaporites. The combined age spectra and K/Ca plots for samples 05-P2, 05-W2, and 06-3.1 exhibit profiles consistent with diffusion kinetics from a single domain and are interpreted to represent the following:

- The oldest step age, ~514 Ma, for sample 06-3.1 is a minimum age for diagenetic precipitation. As polyhalite is

rarely formed as a primary evaporite, we believe this likely represents the age at which backreactions during lithification have precipitated polyhalite.

- The youngest and oldest step ages: ~286 Ma – ~415 Ma for sample 05-W2 and ~292 Ma – ~380 Ma for sample 05-P2 are interpreted to represent a complex mixing age between diagenetic formation in the Cambrian and partial resetting as a result of stress induced recrystallisation. Both the oldest and youngest step ages for samples

05-W2 and 05-P2 occur during the unconformity between the Cambrian Baghanwala Formation and the Permian Tobra Formation (Fig. 2A). Circulating meteoric fluids during this non-depositional time and continuing during

the Permian may have contributed to the dissolution and recrystallisation of the Salt Range Formation evaporites. We believe the complete physical reconstitution of the formation due to extensive deformation during the Cenozoic exhibits a greater control in resetting the K/Ar system and thus is more likely.

Diffusion experiments on two samples from the Salt Range, Pakistan have resulted in closure temperatures between ~254 and ~277 °C at a cooling rate of 10°C/Ma. To our knowledge this is the first experiment to determine a closure temperature for the mineral polyhalite. Due to polyhalite dehydroxylation and its impact on Ar diffusion kinetics these experimentally derived parameters are only semi-quantitative. This is the first study of its type on challenging samples with a complex deformation history; as such, they serve as a first pass on polyhalite diffusion kinetics and a new base for further experiments.

Further groundwork studies on polyhalite's mineral properties in relation to argon diffusivity, ideally with samples having a simpler geological history, would serve to greatly improve our knowledge of this mineral's behaviour and aid interpretation. Parallel step heating and crushing experiments would develop our understanding of diffusion domains in polyhalite and the influence of relict halite generated ($^{38}Ar_{Cl}$) argon. Hydrothermal diffusion experiments on polyhalite make an ideal next step in this work by assessing the diffusion parameters of the mineral at higher temperatures before the thermal decomposition

begins. In particular, discovering whether polyhalite is Ar retentive above its in vacuo dehydroxylation temperature may clarify whether diffusion results purely from this phase change. Similarly, understanding the timing between phase transformation and Ar diffusion in the new crystal structure will be essential in understanding Ar degassing mechanics in both polyhalite and langbeinite. Furthermore, developing our understanding of the link between polyhalite recrystallisation and Ar diffusion is key to the application of Ar geochronology to deformed evaporite units.

The practical use of $^{40}Ar/^{39}Ar$ geochronology on polyhalite is in its infancy. Despite the difficulty of using the mineral as a geochronometer, further understanding of its diffusion kinetics will clarify its potential for use in geological interpretations and develop our understanding of the thermal history of K-bearing evaporite formations.

Author Contributions: Richards, King, and Collins conceptualised the research goals, performed fieldwork including
acquisition of geological samples and initial investigation work. Jourdan performed the formal analysis using established methodology. Richards wrote original and revision manuscript drafts with contributions from all co-authors.

The authors declare that they have no conflict of intertest.

No additional assets (software, models, or data sets) are utilised in this paper that are not supplied in the supplementary data

## 7. Acknowledgements

This work was funded by the Australian Research Council grant #DP120101560. ASC is funded by ARC future fellowship #FT120100340. As well as the Frederick A. Sutton Memorial Grant as part of the American Association of Petroleum Geologists Foundation Grants-In-Aid program. The authors would like to thank the National Centre of Excellence in Geology, University of Peshawar and the Pakistan Academy of Sciences for their support and hospitality as well as the Khewra Mine Deputy Manager of mining Mr Irfam Ahmad and Chief Engineer Mr Bakhtiar Ali for allowing us access and sampling within the Khewra Mine. We are grateful to Marissa Tremblay and two anonymous reviewers for their invaluable input improving our manuscript.

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
