# Peer review of "Deformation recorded in polyhalite from evaporite detachments revealed by $^{40}{\rm Ar}/^{39}{\rm Ar}$ dating"

_Geochronology, 2021_

## Author Response (AR1)

**Reply to Reviewer Comments**

We would like to thank the Marissa Tremblay and both anonymous reviewers for their comments and critiques. We appreciate their efforts in critically assessing our manuscript and believe their comments have greatly improved our paper.

Following is our response to reviewer comments. Initial response shaded purple, post review corrections shaded green.

**Reviewer 1 – Marissa Tremblay**

This manuscript presents $^{40}Ar/^{39}Ar$ step heating data for polyhalite samples from the Salt Range Formation in northern Pakistan, with the overarching goal of using the $^{40}Ar/^{39}Ar$ data to interpret when precipitation, recrystallization, and/or thermal resetting took place in the geologic history of the Salt Range Formation. The motivation for attempting to date the polyhalite is clear, as geochronological constraints for the Salt Range Formation are quite sparse otherwise, and evaporite minerals are generally challenging to date. The $^{40}Ar/^{39}Ar$ data are parsed into two types: age spectra, and step-heating experiments used to derive Ar diffusivities. Both types of data are quite complex, with none of the age spectra yielding age plateaus, and the diffusivities determined yielding complex, curved Arrhenius arrays. I think the authors have done a good job of interpreting (and, importantly, not over-interpreting) the age spectra, inferring only maximum and minimum ages of events from their youngest and oldest step ages, respectively. However, I have many issues with the step heating experiments used to derive diffusion parameters. I outline these methodological and interpretive concerns in my line-by-line comments below, but the summary is that I do not think that have provided quantitative or qualitative constraints on the diffusion kinetics of Ar in polyhalite. Given this, and given the limited geochronological utility of the age spectra, I cannot recommend this manuscript for publication in Geochronology.

Line 48: The Reiners et al. textbook is not really an appropriate reference here. I recommend removing the reference and modifying the second part of the sentence to say, "…several studies have applied $^{40}Ar/^{39}Ar$ dating to evaporite minerals."

Accepted – We will modify this part. Additionally several studies are briefly discussed immediately afterwards and serve to reference this statement. Corrected Lines 47-48

Lines 49-52: This sentence is garbled – please revise.

Accepted – will reword for clarity. Corrected Lines 48-50

Line 53: Please include a reference for the statement about langbeinite being less susceptible to alteration and Ar diffusion.

Accepted – will reference in text Reiners et al. 2018 who writes – "Many studies have been conducted on various K-bearing evaporate minerals, including the pioneering study of Aldrich and Nier [1948], but most are prone to open-system behavior and yield ages that are significantly younger than deposition. An exception is the sulfate langbeinite, which in some cases has been shown to yield plausible deposition ages [Leost et al., 2001; Renne et al., 2001b]." corrected line 53

Line 60: I recommend removing the parenthetical part of this heading; "Geologic Background" is sufficiently descriptive.

Accepted Corrected Line 60

Figures 1 (and 2): There is no legend explaining what the shading means; please include this.

Accepted – the shading in figure 1 references the geological groups/formations in figure 2, denoted with the same shading scale. Clarifying description to be added to the figure description. Corrected Lines 73-74

Lines 138-149: There are many details about the step heating experiments missing here, and it is insufficient to cite the Reiners et al. textbook. For example, how have the authors determined that their raster heating approach achieves persistent uniform heating across all grains? And how have they calibrated the temperature measurement? Is the temperature measurement by pyrometer, and if so how have they calibrated the emissivity of polyhalite to relate measured temperature to true temperature? I am skeptical that the approach described would achieve uniform, steady, well-calibrated temperature measurements, which are all essential characteristics to quantify diffusion kinetics from a step heating experiment. Please quantify the uncertainty in temperature for each heating step, which I presume will be quite large.

Partially agreed: good and very legitimate points made by the reviewer. The issue is that we regretfully omitted to add the correct methodology surrounding the diffusion experiment! 100% our fault, We will now add it in the methodology section. Basically, whereas we degassed several samples with a CO2 laser to attempt to obtain age information with minimum blank levels, we measured the diffusion characteristics of several samples using a temperature-controlled *furnace*. Although furnace don't allow for the maximum age precision due to their larger volume, they allow much better temperature homogeneity and control. Corrected methods rewrite, lines 136 - 149

There also needs to be a citation here for the transformation of gas fractions into diffusivities; e.g., many people use the discretized equation of Fecthig and Kalbitzer (1966), with the modified cut off parameters in McDougall and Harrison (1999). How were uncertainties propagated from gas fraction into diffusivities for use in Isoplot? This is a nontrivial calculation because of the cumulative nature of gas fraction, and either requires a Monte Carlo simulation (e.g., see Tremblay et al., 2014), or the analytical solutions derived by Ginster and Reiners (2018). I recommend recalculating the Arrhenius plots using the Excel spreadsheets provided in the supplement of Ginster and Reiners (2018), as this will also allow the authors to put in uncertainties in the temperature.

Partially agreed: we will now report a range of possible calculation for the Ea and E0 and their uncertainties in the text and provide much more details in the methodology and results. In more details:

Another interesting discussion point by the reviewer. We used standard equations from McDougall and Harrison (1999) for calculating the D0 and Ea where ultimately, the errors for Ea and D0 arise from the dispersion of the data along the regression line calculated with Isoplot and are not input from the steps uncertainties. The reason behind that is simply that the dispersion, intra-sample but and certainly and especially extra-sample is usually significantly higher than the uncertainties of

the measurements hence the extra level of calculation is not warranted. Furthermore, as described in Thern et al. (2020), the final uncertainties from D0 and Ea from multi samples can propagated by Monte Carlo simulation by combining results and their dispersion uncertainty in a single average number.

Another important point is that in our original manuscript, due to the calculation were initially performed, we unfortunately used an old version of our diffusion spreadsheet that was corrected in 2018 following a reviewer's feedbacks and successfully implemented by Blereau et al., 2019 and Thern et al., 2020. We will now be using the correct spreadsheet and as a result, our results varies slightly compared to our original submission. So thanks to the reviewer for indirectly make us look at our results as we are happy to have been able to catch that before this study is published. Consequently, we note that our calculation for the 1/T and LnD for each step yield the exact same results as in the Spreadsheet of Ginster and Reiners, so the only difference comes from the regression of the 1/T vs LnD space and associated error propagation.

In essence, the spreadsheet for Ginster and Reiners (2018) proposed a range of solution and error propagation. One of them is the one we used in this study and is the non-weighted ordinary least-square regression and therefore use non-weigted solution which does not need uncertainty to be calculated. We calculated using such a function in isoplot. That approach allows calculating the Do and Ea uncertainties that arise from the deviation of the slope. As stated by these authors, and quite intuitively, there should be no difference between the models if the errors are similar across all steps used in the calculation. We did test the difference on 5 data points on our most robust sample, we also used our own calculation regressed in Isoplot in comparison. Slight differences are observed in between different approaches. Note that Isoplot is a tool used by the entire geochronology community and which has proven extremely robust for its calculation, particularly for its regression tools. All results are at 95% conf level.

Isoplot: Ea = 260 (+2 / -79)  /  D0 = 2.71 (+2.69 – 2.70) E6  /  Tc ~ 254 °C

OLS (excel): Ea = 244 (± 20)  /  D0 = 188 (± 588) E5 /  Tc ~ 244 °C

Weighted LS: 260 (± 2)  /  D0 = 2.97 (± 2.10) E6 / TC ~ 252 °C

Our approach therefore seems very conservative in term of uncertainties compared to the one listed in Ginster and Reiners (2018) yet, the values are very similar. We will now report all 3 sets of values for the reader but we stressed that due to all the geological uncertainties, our conservative approach is probably the more realistic. Corrected methods rewrite lines 159-169

*"Due to polyhalite dehydroxilation and its impact on Ar diffusion kinetics these experimentally derived parameters are only semi-quantitative and must be taken with a grain of salt. As such, they serve only as a first pass on polyhalite diffusion kinetics and cannot be used for geochronological works with any precision."*

This aspect is critical and we will attempt to do a better job portraying this message in the discussion itself.

Finally, how did you determine the diffusion domain/grain size? Were grain dimensions measured via microscopy? Are there any concerns about changing the diffusion domain size when the sample was crushed to obtain the polyhalite separates?

The grain size was associated with the standard grain size for sample preparation obtained by sieving. Multi-particles were pooled together for ensuring enough gas. The grain sizes (diameter) varies from 150-210 µm which is what we used in our calculation and error propagation and with an average radius of 90 µm. Corrected lines 122-124

Line 151: Please change to "$^{40}Ar/^{39}Ar$ dating"; it's redundant to say "age dating." Please make similar changes throughout the text (e.g., Table 1).

Accepted – will remove the word "age" in this context throughout the text. Corrected Lines 47, 48, 171, 173, 176, 341

Line 155: In this sentence and elsewhere, the use of the word apparent is unnecessary. The age spectra are not apparent, they are the observed age spectra and the measured step ages.

Accepted – will reword appropriate sentences with "observed" and "measured" in place of apparent. Corrected Lines "observed": 182, 215, 222 382. "measured": 19-21, 176, 183, 192-193, 195, 212, 219, 281, 293, 298, 303, 311,

Lines 162-164: Please add citations for the plateau criteria.

Accepted - Will add (Jourdan et al., 2020). **F. Jourdan**, T. Kennedy, G.K. Benedix, E. Eroglu, C. Mayer. Timing of the magmatic activity and upper crustal cooling of differentiated asteroid 4 Vesta. *Geochimica Cosmochemica Acta 273, 205-225. 2020.* Corrected Line 184

Lines 177-178: A flat K/Ca spectrum does not imply argon diffusion from a single diffusion domain. If the diffusion domain size. If a mineral of uniform composition is comprised of multiple diffusion domains and/or multiple grain sizes, the K/Ca spectrum will be flat. The authors have not made a convincing argument as to why the grain size represents the diffusion domain for polyhalite. I recommend this sentence and similar statements made elsewhere be removed.

Partially accepted - An homogenous K/Ca does not show that there is only one "domain" but rather than the sample is compositionally homogenous. Note however, that this is an important point and a good start. In order to derive reliable diffusion parameters, it's best to derive it from homogenous single-domain mineral (in fact, mineral with multi-domain is suspicious to start with, especially on why there are several domains to start with (cracks, alteration, etc…). what is more convincing though, is the alignment of points in the Arrhenius plot. Here, we derive the diffusion data from the dominant array as we suspect, the curvature is due to dehydroxyaltion break down. We will make that clearer in the text. Added extensively to the discussion, corrected lines 357-366

Line 186: Delete the words "quality data is not considered presentable."

Accepted – will remove this part of the sentence. Removed line 186

Lines 220-230: Like the methods section, there is a lot of detail that has been left out here in the description of the step heating experiment results, and. Why are only the first three steps and four steps utilized for the linear regression on experiments SRL05 and SRL06, respectively? What was the criteria for choosing which steps to fit? None of the behavior beyond the first few steps

is discussed, despite a lengthy discussion on the structural transformation of polyhalite beginning at temperatures below those traversed during the step heating experiments. Why does the Arrhenius plot become curved, and why is there a giant spike in diffusivity around 500 ºC? The temperature labels on the top x-axis of Figure 7 are incorrect, and the bottom x-axis needs units.

Accepted - we will add much more details in the text. This part was arguably rushed. Comments regarding the Arrhenius plots are covered a few comments earlier and will be incorporated in an updated manuscript. Results section has been completely rewritten with updated figures. Lines 170 - 268

Table 2: This table is unnecessary; I recommend simply discussing existing diffusion parameters for other minerals in the text.

Accepted - Table removed. Table. 2 is now a results table for the step-heating experiment lines 266 - 268

Lines 264-301: This is an excellent discussion of the structural transformations that polyhalite undergoes at various temperatures. Unfortunately, as the authors point out, all of their heating steps in their step heating experiments occur at temperatures above the temperatures at which these structural transformations occur. So I think describing the step heating experiments as a semi-quantitative measure of the Ar diffusion kinetics in polyhalite is unwarranted. Instead, their step heating experiments most likely document the release of Ar from two langbeinite-type phases, per their eq. 3. This likely explains the complex Arrhenius behavior the authors observed. I recommend adding discussion to this effect.

Partially accepted – we agree that the step heating experiment likely documents the release of Ar from more than polyhalite as it most likely includes the two langbeinite-type phases and the higher temperature (>646˚C) triple salt ($K_2CaMg (SO_4)_3$ among other possible transformations at higher temperatures. We agree that adding a section discussing these transformations as explaining the complex Arrhenius behaviour will improve our interpretations and discussion section.

Since there is so little information regarding polyhalite and no established minimum closure temperature (excluding our results) we believe our description of results as semi-quantitive for polyhalite Ar diffusion kinetics is warranted. it is the first step in a chain of mineral transformations. While the first dehydration reaction of polyhalite onsets at 237˚C and peak dehydration occuring at 343˚C it does not imply total phase transformation at this temperature. 3 hypotheses can be made: 1, polyhalite releases Ar before and during transitioning to and the langbeinite-type phases as it continues to degas as the mineral Tc is unknown. 2, all argon release is due to phase transformation and polyhalite is Ar-retentive well above its mineral decomposition temperature. 3, if our calculated Tc are considered it indicates a mixture of Ar release from both polyhalite and langbeinite phases during transformation. Section 5.4 added to the discussion regarding this topic, lines 406 – 429.

Lines 333-341: This is a nice discussion of the potential effects of deformation on recrystallization and Ar diffusion. However, if intracrystalline domains are known to be present in these samples, that seems to contradict repeated statements in the text that Ar diffusion

from their samples occurs from a single domain represented by the grain size. If anything, this is a compelling observation to expect the presence of complex, multi-diffusion domain behavior.

Accepted - we agree, this effect and phase changes is why we put all the limitation on the interpretation of the diffusion parameters and how to use those data. It is also why using the model spreadsheet proposed by Ginter and Reiners yield results that are way too precise compared to the real (geological) uncertainty (by opposition to analytical uncertainties. We therefore warn the reader of all the potential caveats – yet, knowning a ball park for Ea, D0 and Tc is nice and will help people make interpretation in case diffusion is the issue. Updated the discussion to address the diffusion domain (lines 356-366)

Lines 356-366: Given the dubious geologic significance of the oldest and youngest step ages, I recommend against quoting the step age uncertainties here. Instead, I suggest that you quote approximate ages, e.g., "~514 Ma" rather than "514 ± 3 Ma" in the first sentence.

Accepted – will remove age uncertainties and replace with ~ approximate ages, we are not sure why this was not caught by us earlier. Corrected all instances including data in table. 1

Supplementary files: Again, I urge the authors to use the supplemental spreadsheets of Ginster and Reiners (2018) so that they can propagate uncertainty in temperature and gas release fraction into the calculated diffusivities and therefore their linear regressions.

Accepted: Cf. answer above – New supplementary data sheet created.

**Reviewer 2 – Anonymous**

**General comments:**

This manuscript reports Ar/Ar step-heating results from polyhalite in the Salt Range Formation in northern Pakistan. The authors aim to provide geochronologic age constraints for this formation (on the timing of precipitation, deformation, and/or thermal resetting), and to determine diffusion parameters for polyhalite. Polyhalite is a potassium-bearing salt [$K_2Ca_2Mg(SO_4)_4 \bullet 2H_2O$], formed in evaporite environments, so is an interesting target for Ar/Ar geochronology, especially as this phase has only rarely been analysed via this technique. Unfortunately, it appears that the samples chosen for this study have experienced a complicated geological history of deformation and reheating, with none of the samples yielding Ar/Ar age plateaus. The Arrhenius diffusion results are also non-linear, and likely confounded by dehydration and structural reconfiguration reactions, complicating attempts to calculate diffusion parameters for polyhalite.

The results of this study could possibly be suitable for publication in Geochronology, as even less than ideal analyses may potentially be useful when reporting results for a little-studied mineral or region. However, this usefulness would require the results and discussions to be conservative, noting the pitfalls in the data in a consistent way, and framed to highlight aspects that could lead to improvements. This manuscript has several crucial points that require resolution before it would be suitable for publication.

**Specific comments:**

The most important points that need improvement are as follows:

Line 19: states 'the established early Cambrian age of the formation'. However, Figure 2 indicates that the that the Salt Range Formation is Ediacaran-Early Cambrian. If no other geological constraints are available, the upper limit for the age of the formation is 635 Ma (the base of the Ediacaran) - not 541 Ma (the base of the Cambrian).

Uncertain – the Salt Range Formation is known to have been deposited during the early Cambrian as it contacts the overlaying Khewra Formation which has been dated in the early Cambrian. To date there has been no successful age dating of the Formation itself, in part a reason for this work. Using the 541 Ma age is therefore a more accurate formation age estimate than the base of the Ediacaran at 635 Ma. No change made

The methods section currently lack sufficient details about the diffusion experiments. The methods section (around line 138) does not detail the technique used to measure the step temperature, which is crucial for diffusion experiments. The methods section must therefore be expanded to explain the methodology for the diffusion experiments and how the diffusion parameters were calculated.

Accepted – will be updated following reply to reviewer 1 comments. methods rewritten, lines 118 - 169

Line 150-218: the entire section for the step-heating results needs improvement. The current separation of the age spectra and results section into Figures 4 (the 'good' data) and Figure 5 ('erroneous data') is confusing, as some of the samples currently in Figure 5 (06-2.1, 06-3.2) have similar spectra to those in Figure 4. I would recommend splitting the results into 1) aliquots with high K/Ca (i.e., separates of polyhalite), and 2) aliquots with low K/Ca, imprecise step ages, and uninterpretable results. Crucially, for samples with low K/Ca, this observation indicates that the phase sampled and analysed was likely not polyhalite but is instead another mineral with lower potassium.

Accepted – our initial desire was to post all the results due to the limited data available on polyhalite and splitting the results between interpretable and erroneous made more sense for the reader to more easily identify the results discussed in the text. Separating the results based on K/Ca values may lead to more confusion unless the text is heavily reworded. Though this can be done to incorporate other reviewer comments. Additionally, the observation that samples with low K/Ca values are likely not polyhalite and so have imprecise step ages is worth adding to the discussion. methods rewritten, lines 118 – 169. Additionally figures 4-6 have been rearranged and reorganized following reviewers suggestion to separate low and high K/Ca value samples.

Line 150-218: in a revised results section, please reduce the amount of duplicated information between paragraphs and between the main text and tables. Also ensure the results are not over interpreted - caution should be applied in attributing geological significance to the youngest and oldest steps from a disturbed heating spectrum, particularly if there is no reproducibility between aliquots. A conservative interpretation of the data presented in this manuscript indicates: 1) That the Salt Range Formation is likely older than ~500 Ma (i.e., the oldest step with good precision (Fig 6). 2) That the polyhalite Ar/Ar results yielded a broad range in ages from ~500 to 200 Ma, likely due to variable and incomplete resetting of the polyhalite via tectonic and thermal events in the region. 3) That the most recent geological event in the region that affected the polyhalite occurred less than ~200 Ma (based on the youngest step with good precision (Fig. 6)). However, if the youngest step was only partially reset by that geological event

(or events), the event could be much younger than 200 Ma - especially given that tectonic activity in the Himalayas is occurring in the modern day.

*Accepted – a revised results section incorporating other reviewer comments will be streamlined to reduced unnecessarily duplicated information. We have endeavoured to not over interpret the data and believe our interpretations as presented reflect an accurate representation of the data. Corrected methods rewrite, lines 136 - 149*

Line 153: states that 'Polyhalite single crystals, polycrystals and grain aggregates taken from larger samples underwent step-heating 40Ar/39Ar age dating.' However, from the information provided it is not clear which samples were single crystals, polycrystals, or aggregates. This information must be provided somewhere (e.g., in Table 1) for each of the aliquots analysed. Also, please briefly explain the difference between a polycrystal and a grain aggregate.

*Accepted: single crystal were analysed for laser heating, whereas multicrystals aliquots were used for diffusion experiments with the furnace. This will be made clearer in the text. Corrected lines 153-154*

The results of the diffusion experiments (lines 220-226) are far too brief. This section only spans six lines, and this text is not useful, only containing a series of numbers already presented on Figure 7. The authors need to explain the choice of samples used for the diffusion experiments, outline how the diffusion parameters were calculated, and describe the features seen on the diffusion diagrams (e.g., slope, slope changes, spike at 500°C).

*Accepted – we will drastically expand this section (following the approach some of us used in Blereau et al. (2019) and Thern et al. (2020). Corrected methods rewrite, lines 136 – 169, and presented/discussed in the rewritten results section, lines 193 – 233.*

As this study represents analyses of a relatively unstudied mineral (polyhalite), it could be beneficial to have a clear list of what worked, what didn't work, and what could be improved for future analyses. There are aspects of this study that were successful include 1) that some samples yielded high and consistent K/Ca values, indicating the aliquots analysed had a consistent mineralogy; and 2) that the samples had high percentage of radiogenic argon (i.e., that polyhalite is reasonably good at keeping the Earth's atmosphere out of its mineral structure). Both are basic – but non-trivial – observations.

*Accepted – part of the conclusions section does make note of possible future work but would benefit from including a short commentary of the successful and unsuccessful aspects of this work and expand upon the areas for future improvements. Added to the discussion, lines 358 – 363, and amended the conclusions, lines 454 - 463*

It may also be useful to have some recommendations for future work, including further groundwork studies on polyhalite. Presumably such groundwork studies could be more easily done in areas with a simpler geological history.

*Accepted – adding a short section in the conclusions recommending future groundwork studies will improve the paper. As above the conclusions section was amended, lines 454 - 463*

In the documents supplied for review, I was unable to find a table containing the analytical results, which is crucial for reporting Ar/Ar data. Please ensure such a table is incorporated

(likely as a supplementary dataset), and that it includes all information required (see Renne et al. 2009 Data reporting norms for 40Ar/39Ar geochronology. Quaternary Geochronology v4 p346-352).

Disagree – the dataset that we submitted with the manuscript should have contained the entirety of the raw data for each sample. Individual excel spreadsheets with multiple tabs contain all the data used for these experiments. New supplementary spreadsheet created

Several areas of text are poorly written or confusing. Please go over the full manuscript and ensure clarity for both general writing and scientific concepts. Some particularly notable examples include lines 14-18, 24, 62-63, 130, 303-304, 315-318, 326-327 (which has quite a jump in context between line 326 (talking about processes in the Permian) and 327 (talking about modern day processes), 359-366, 371.

Partially Accepted – some of the examples presented can be rewritten for clarity however many of these examples are merely concisely worded but grammatically and technically accurate. Where appropriate these will be reworded for clarity. The discussion jump between Permian and modern processes has been clarified to directly state both as possible scenarios for clarity, line 323 & 328-329. Other stated examples are in sections that have also been rewritten as part of the methods/results update.

**Technical corrections:**

Line 30: write the chemical formula for halite – all the other minerals in this portion of the text have formulae.

Accepted corrected line 30

Line 35: start a new paragraph.

Accepted corrected line 35

Line 59: '…the deformation history'.

Accepted corrected line 59

Figure 1: label the x and y axes as 'Longitude (°E)' and 'Latitude (°N)'.

Accepted – will add axis values to the map. This map is an already published figure (Fig. 1 in Richards et al., 2015) as such redrafting is unnecessary so no change is made.

Figure 1: in the caption, mention that the study site is the Kewera mine. On the figure, can the authors write 'Kewera' in a different font (e.g., red)? This will be useful to draw the reader to the site.

Uncertain – seems unnecessary – No change made.

Lines 88-102: this text is essentially a duplication of information in Figure 2. The main text would be streamlined and improved by removing this text or moving it to the figure caption.

Rejected – this section is a succinct description of the stratigraphy and is essential to properly contextualise the geological background. While there is some overlap (particularly in the description of the Salt Range Members) there is a difference in the information presented. No change made.

Lines 106-111: the minerals described are not identifiable in the hand-sample images of Figure 3A and 3B. Could the authors also supply other images that identify these minerals if available e.g., close-up photos/thin section photos/SEM.

Accepted: it will be clarified that these samples are the same as published in Richards et al., 2015, which includes a detailed description and thin section analyses. Adding an additional line to the figure caption stating so and directing the reader to this publication should allow for sufficient identification of mineralogy without overloading the reader with additional figures and images. Corrected figured captions updated, lines 114-116

Figure 3 caption: extra text is required to explain what can be seen in these images. Also, please specify the scale.

Accepted – the description of these samples is in the text but will be added here for clarity Corrected figure captions updated, lines 114-116

Lines 115-138 (and elsewhere in the manuscript): please ensure correct use of superscripts and subscripts.

Accepted – this has been triple checked, all subscript & superscript should be accurate.

Line 115: what was the grainsize of the crushed aliquots?

Accepted - we initially used the minimum grain radius of 75 µm but have now updated this value to 90 ± 15 µm to encompass the range of possible values after sieving (diameter between 150 and 210 µm). Added lines 122 – 123.

Line 118: replace 'alteration' with 'dissolve'.

Accepted corrected line 119

Line 124: J factors are reported with too many digits and should be limited to significant figures only.

Partially accepted - We would tend to agree for reults (e.g age), but here, this is a parameter and to allow the exact same calculation, providing all these parameters are they used in the initial claculation spreadsheet seems like a good idea. All J-values for each sample analysis is presented in the supplementary spreadsheet

Line 124: which J factors correspond to what samples?

Accepted – these can be found in the raw ArCALC sheet supplied in Annex - All J-values for each sample analysis is presented in the supplementary spreadsheet

Line 126: what was the frequency of air pipettes? How many air pipette analyses were included in the discrimination calculation?

Accepted - Air shots are measured every 2-3 samples and are average on a time period basis as they don't vary. That information is not necessary in the text as it does not change anything as long as the discrimination is well measured. – no change made.

Line 129: write the abbreviation 'IR' out in full.

Accepted - will replace with 'Infrared' corrected line 135

Line 155-157: delete the following text as it is poorly written and unnecessary in the main text. 'Apparent age spectra plots display the apparent ages for each step of the experiment and are calculated representing a percentage of cumulative 39Ar released with the last step resulting in 100% 39Ar released from the sample. Stacked below each age plot are the K/Ca ratios.'

Accepted lines removed 155-157

Line 162-164: provide a reference for this definition of a plateau.

Accepted - Jourdan et al. (2020) as requested by reviewer 1.. added line 184

Lines 175-179: this information should be in the figure caption.

Accepted – will move this section to the figure caption. Results section entirely rewritten, lines 170 – 268, with updated figure captions for the rearranged figures 4-6.

Line 188-198: suggest changing to 'Sample 06-1.2 produced very little gas, with low K/Ca values and analyses barely above blank levels, yielding very imprecise ages; the results from this sample have therefore been discarded.

Partially Accepted – we agree with changing this sentence however we believe the reviewer intended this to apply to lines 188 – 192 as the following two sentences in the paragraph cover more than just this sample. Results sections has been rewritten, lines changed 210-211.

Table 1: as none of the samples yielded useable plateau or isochron ages, the following columns should be deleted: Plateau Age, ±, MSWD, %39Ar, Inverse Isochron, MSWD. Deleting these columns will free up space in the table.

Accepted – normally including this additional detail is considered standard for reporting however in line with reviewer 1's comments "I recommend against quoting the step age uncertainties here" removing these columns will certainly free up space and improve readability. An additional comment will be added to the figure caption stating the reason for not presenting Plateau Age, ±, MSWD, %39Ar, Inverse Isochron, MSWD columns. These columns have been removed and comments added to the table caption, lines 176 – 179.

Line 248: delete 'closed'. If the system has been partially or wholly reset, it is not closed.

Accepted. Corrected line 286.

Table 2: numbers are reported with too many digits – limit to significant figures only.

Accepted – the values here are taken directly from their publications and presented at 2 decimal places. The table will be removed and parts added to the text per other reviewer comments. Table removed and replaced with step-heating diffusion calculation table, lines 266 – 267.

Line 277: what are the 'various analogues'?

Uncertain – the polyhalite analogues are ployhalite crystals with various cation replacements (Mn, Fe, Co, Ni, Zn & Leightonite) and are presented in Wollman et al., 2008 as mentioned in the text. They are mentioned here as the individual analogues have wildly varying thermal dehydration onset temperatures (185-311 ˚C). We believe a full description of their thermal dehydration characteristics will unnecessarily bloat out the text and derail the readers thought train so was not included. Analogues listed in line 380.

Line 289: what is meant by 'with variations to pressure'?

Uncertain – some thermal experiments are not run at atmospheric pressure but stating all the pressure variations used for these experiments is unnecessary. No change needed.

Line 300: please explain why that approach was not possible in this study.

Accepted – these experiments were completed as part of a PhD project and the facilities were not available for a hydrothermal experiment during this time. Changed line 404.

Line 315: instead of 'magmatic' do the authors mean 'sedimentary'?

Rejected – this sentence focuses on the difference between the sedimentary/chemical precipitate nature of polyhalite and the magmatic nature of most other minerals used for Ar/Ar dating. We believe it has been used correctly in this context. No change needed

Line 356: here and elsewhere: this should be 'minimum age'.

Accepted – similar to the first reviewer's comments, this will be changed throughout the text. Corrected line 436, all other instances were correct.

**Reviewer 3 – Anonymous**

**General Comments:** This study applies the $^{40}Ar$-$^{39}Ar$ dating technique to crystals of polyhalite from the Salt Range Formation in northern Pakistan. The aim is to determine the age of the formation and to determine diffusion parameters and a closure temperature for Ar in polyhalite. The study did not yield any crystallization ages due to disturbances to the Ar-Ar system, suggested to be from repeated episodes and deformation (as well as the difficulty associated with dating evaporite minerals in general). The authors were able however, to present minimum ages suggested to represent primary crystallization, and a suggested maximum age of deformation. There are several issues with the study that require clarification, further information and revision (see below), but study is still a useful stepping stone, particularly because of the paucity of geochronologic data for evaporite minerals in general. Clarification on the determination, and meaning of, the reported range of closure temperatures determined for polyhalite is required. In the introduction, the authors are suitably cautious "Though these results are semi-quantitative, they are contextualised with the structural history of the host formation to form a speculative interpretation of deformation history" and generally I think this is fair. As reliable Ar-Ar ages for polyhalite would be useful for, e.g., formation ages, reconstructing deformation histories, the qualitative to partially quantitative data provided in this study would likely be of use to future workers pursuing Ar-Ar applications of these complex minerals. As the results are largely qualitative, however, the discussion could be shortened in places.

**Specific Comments:**

Line 132: Was the "homogenised temperature" determined – was it specifically monitored?

Accepted - This sentence refers to the laser step heating experiment and jogging the laser beam around the sample that has an homogeneization effect in case the laser beam is heterogenous in power. As discussed above, we will add information regarding the diffusion experiments performed with the furnace where, in this case and by design, the temperature is homogenous. Corrected methods rewrite, lines 118 - 149

Line 138: It is mentioned that "contemporaneous step heating experiments" using the same parameters as described in the analytical methods were used to determine diffusion parameters. Much more information on these experiments is needed here. It would be very helpful to describe (as above) how the temperature was monitored, e.g, pyrometer?, as this is essential for these experiments.

Accepted - The temperature of the furnace was calibrated electronically as per Pond engineering recommendation. Corrected methods rewrite, lines 136 - 169

Line 153: You describe 9 samples but only descriptions for SRLR-05 and SRLR-06 are given in section 2.2. If the 9 samples are aliquots of these two samples, perhaps say nine aliquots or repats from two samples. Looking at the filenames in the supplement, it looks like 7 from 05 and 2 from 06. It might be helpful to list the number (e.g., n=7) for each sample and also explain the uneven distribution (why only 2 analyses from sample 05?).

Accepted – there are only two samples taken from Khewra mine (SRLR-05 & SRLR-06) from these, 9 smaller samples (aliquots) were separated for analysis. SRLR-5 (n=2), SRLR-06 (n=7). There is a

strong compositional difference between the two samples with SRLR-05 comprised of 95% Halite so finding polyhalite crystals was difficult, hence only 2 smaller samples. Will reword the text for clarity to include this information. Corrected lines 172 - 174

Line 155: If you denoted each panel with an (a) or (b) the description of data plots here would be less cumbersome.

Accepted – will include (a) & (b) icons for clarity describing the age and K/Ca plots. Figures 5 & 6 have been updated with (a), (b), (c) etc icons.

Line 158: (Figure 5) If the data aren't used for anything, I would suggest moving these plots to the supplement.

Uncertain – As so little data is presented in the literature on polyhalite we believe having all our data in the paper holds some value. Additionally, it helps display the difficulty of dating polyhalite which ties in with the planned improvements to the discussion/conclusions relating to what did and did not work well. Figures 4-6 have been reworked according to previous reviewers comments splitting age & K/Ca plots by K/Ca values. Lines 191, 206, 225

Line 178: Can you explain why the flat K/Ca indicates a single Ar domain?

Accepted - It does not, but it's encouraging in this direction. Cf. answer to reviewer 1 with the same question and will be clarified in the text. Section added to discussion, lines 356 - 366

Line 185-187: Long sentence – a bit unclear, please revise. Also, seems more interpretative rather than "results."

Partially accepted – this sentence is located here to premise the use of presenting & interpreting poor data. Typically, poor quality data is not presented but we have done so here due to the scarcity of polyhalite age data. Will reword for clarity and incorporate other reviewer comments for this section. Lines removed and rewritten as part of results rewrite

Looking at Figure 4 and 5, two samples of SRLR-06 have identical K/Ca to SRLR-05 of ~ 0.82 but the remaining 5 aliquots of SRLR-06 are entirely different. Can you shed light on the two populations?

Accepted – this was raised by the 2nd reviewer who suggests that low K/Ca values could indicate the sample is not polyhalite. This will be incorporated into the discussion. Added to results section, lines 222 – 224.

Line 242: It still isn't clear how single domain diffusion was established – I might be missing something, so please expand/clarify.

Accepted - Cf. answer to reviewer 1 above. Section added to discussion, lines 356 - 366

Line 296: If the polyhalite is undergoing dehydroxilation at the same T range as the determined closure temperature, then much of this discussion could be shortened and perhaps out of caution the closure temperature shouldn't be reported in the abstract, or with a caveat.

Uncertain – We believe the description of our closure temperatures and diffusion kinetics results as semi-quantitative accurately reflects the value of these results. As reviewer 1 pointed out our experiment is likely measuring two langbeinite-type phases and the higher temperature (>646˚C) triple salt ($K_2CaMg (SO_4)_3$ however as the response to that comment also covers, we don't know at which point we are measuring polyhalite, langbeinite, or the triple salt. As such we believe keeping our current description of results is warranted and adding further to the discussion is more valuable than dismissing the results. This has been added as section 5.4 to the discussion.

Line 315: Perhaps step-crushing experiments alongside step-heating experiments could help here.

Accepted – yest it certainly might help, but definitely not available at our facility nor commonly available in many Ar labs unfortunately. Incorporated as line 456

Line 324-330: Are you suggesting this? It isn't clear the way it is written. This section seems quite speculative.

Accepted – we believe it is a possible explanation, though not the most likely, for the results we have observed and thought it worth mentioning. Will reword for clarity. Reworded for clarity, lines 328 - 329

Section 5.1/5.2 and section 5.3 could be reversed with the discussion/interpretation of the ages first.

Accepted – Sections 5.1 – 5.3 reorganized and section 5.4 added.

Section 5.1: Relating to the comment above regarding T steps, more information is required here. Also, its quite short to warrant its own section as it currently stands.

Accepted – we will expand this section to incorporate a few different reviewer suggestions as discussed above. This section is heavily expanded as part of the discussion restructure, lines 349 - 366

Line 333 onwards: The numbered points are confusing – are these possible scenarios, a suggested sequence, etc.? It also seems speculative given the nature of the data.

Uncertain – this section is speculative based on our experimental results, mechanisms of K/Ar diffusion, and geological context. we have offered a few scenarios that may occur individually or simultaneously. An updated section will make the premise for these scenarios clearer. Reworded to define possible scenarios, lines 328 - 329

Line 351: I think 9 samples is a bit misleading, rephrase as above.

Accepted – similar to previous comments and will be updated. Rephrased line 172

Line 354: "profiles consistent with pure diffusion kinetics from a single domain" I don't think this has been adequately demonstrated. I'm not sure that attributing a specific process to the ~ 500 Ma age is warranted.

Accepted - it is consistent with, but one could obtain a similar profile from recrystallization. What we are trying to portray is that diffusion cannot be ruled out for this process. Section on diffusion domain added, lines 356 - 366

Line 370: I would delete "taken with a grain of salt."

Rejected – a universal expression used to warn the reader that these results should not simply be taken as is. In addition, it is thematically topical and relevant to both the results and nature of the samples. Removed after further revisions.

Line 371: "As such, they serve only as a first pass on polyhalite diffusion kinetics and cannot be used for geochronological works with any precision." It seems this sentence negates all prior discussion. I think with different language you could frame this in a more positive light –it is a first study on very challenging samples and much can be learned from the experiments.

Accepted – this is a nice way to frame the results, will update this sentence. Corrected lines 450 - 453

Figure 6 could do with revision. The fonts are too small, there are excessive decimal places in the y-axis, the legend is too small, and most of the data are in at the axis (could do with a zoomed in panel). The symbol colors of 06-1.2 and 06-2.1 are hard to differentiate. I would also remove the title.

Accepted – will remake this figure for clarity including this and previous comments. Figures 6 has been remade as figure 4 and adjusted for readability and clarity, lines 191-192

Figure 7 would also benefit from revision. The two panels have different scales on both x and y making it difficult to compare visually. The diffusion parameters overlap with the data in panel A. I would also remove the titles.

Accepted – will revise the figure for clarity and simplicity. Figure 7 has been remade and now easier to read, line 238.

The formatting of Table 1 makes it really difficult to read, perhaps adjust columns or delete "incalculable" and replace with "nd" for not determined.

Accepted – as suggested by reviewer 1 (removing age uncertainties to reflect the quality of data) and reviewer 2 (removing the columns Plateau Age, ±, MSWD, %39Ar, Inverse Isochron, MSWD) will be incorporated and make table 1 more readable. Table 1 revised, line 176.

Table 2 is a nice summary, but it isn't exhaustive and many of the mineral phases are not particularly relevant to the geologic with discussion.

Accepted – the table is not intended to be a complete list of relevant minerals, its purpose is to frame the results and give the reader a comparison with commonly dated minerals. Table will be removed and discussed in the text instead. Table 2 removed and replaced with step-heating diffusion calculations, line 266.

Supplement "Arrhenius Final" could be in better shape for a supplementary file. I can see that the required information is there, but it is quite hard to follow. It also isn't clear how some of the important parameters were determined (e.g., T). This requires clarification and improvement in the main text and in the supplement.

Accepted – the supplementary files will be updated with the newest/corrected version of the data files and made more easily readable. New supplementary file created.

The references need some attention. They are inconsistent in formatting, contain many typos and often have incomplete information.

Partially accepted – Bibliography will be reviewed - All references have been checked for formatting and DOIs added where possible.

**Technical Corrections:** A few minor suggestions to improve clarity and formatting.

All accepted

- Line 50: could delete radioisotope corrected
- Line 56: could delete "process in an attempt" corrected
- Line 60: I would remove (stratigraphy, composition1, samples) from the Header corrected
- Line 116: Superscript 38Ar corrected
- Line 120: Space, neutron fluence corrected
- Line 131: Argon corrected
- Line 132: Superscript -14, also line 136 corrected
- Line 240: "Coincidentally these samples coincide" is a bit awkward, perhaps revise corrected
- Line 254: I would use different notation here, e.g., $2.62 \times 10^7$ corrected
- Line 256: "Ar datable minerals" sounds a bit awkward, perhaps revise. corrected
- Section 4.2, lines 220, would suggest not using "E" notation in the main text. corrected
- Line 276: delete repeated degree symbol corrected
- Line 338: "Deformation events resulting in deformation" could be rewritten corrected